# Structural adaptation of fungal cell wall in hypersaline environment

Liyanage D. Fernando[1,7], Yordanis Pérez-Llano [2],
Malitha C. Dickwella Widanage[1,8], Anand Jacob[1], Liliana Martínez-Ávila [2],
Andrew S. Lipton [3], Nina Gunde-Cimerman [4], Jean-Paul Latgé[5,6],
Ramón Alberto Batista-García [2] ✉ & Tuo Wang [1] ✉

Halophilic fungi thrive in hypersaline habitats and face a range of extreme conditions. These fungal species have gained considerable attention due to their potential applications in harsh industrial processes, such as bioremediation and fermentation under unfavorable conditions of hypersalinity, low water activity, and extreme pH. However, the role of the cell wall in surviving these environmental conditions remains unclear. Here we employ solid-state NMR spectroscopy to compare the cell wall architecture of *Aspergillus sydowii* across salinity gradients. Analyses of intact cells reveal that *A. sydowii* cell walls contain a rigid core comprising chitin, β-glucan, and chitosan, shielded by a surface shell composed of galactomannan and galactosaminogalactan. When exposed to hypersaline conditions, *A. sydowii* enhances chitin biosynthesis and incorporates α-glucan to create thick, stiff, and hydrophobic cell walls. Such structural rearrangements enable the fungus to adapt to both hypersaline and salt-deprived conditions, providing a robust mechanism for withstanding external stress. These molecular principles can aid in the optimization of halophilic strains for biotechnology applications.

Extremophiles are organisms that survive and thrive in harsh environments characterized by unfavorable temperature, pressure, acidity, and salinity[1,2]. Understanding their adaptation strategies can gain insights into the origin of life under extreme conditions and provide solutions to geo-ecological challenges[3–5]. Halophilic and halotolerant fungi inhabit hypersaline habitats and have shown their potential in various industrial applications, such as contaminant treatment of saline wastewater, fermentation-based production of high-value molecules and pharmaceuticals, and biofuel production[6–8]. Halophilic fungi also hold promise as a source of transgenes encoding for salt-tolerant proteins to enhance the halotolerance of other organisms[7,9]. These applications have not

reached their full potential due to our incomplete understanding of adaptation mechanisms.

When exposed to hypersaline environment, fungi need to maintain positive cell turgor pressure. This requires a multitude of cellular processes, including the accumulation of compatible organic solutes, modification of cell membrane composition and fluidity, pigment production, ion homeostasis, as well as cell wall remodeling[10,11]. These physiological responses involve changes in gene expression profiles to provide osmotic balance, oxidative stress management, and metabolic rewiring of the fungal cells[11,12]. Morphological changes have also been observed in the cell walls of the model basidiomycetous halophile *Wallemia ichthyophaga* and the extremely halotolerant black yeast

[1]Department of Chemistry, Michigan State University, East Lansing, MI, USA. [2]Centro de Investigación en Dinámica Celular, Universidad Autónoma del Estado de Morelos, Cuernavaca, Mexico. [3]Environmental Molecular Sciences Laboratory, Pacific Northwest National Laboratory, Richland, WA, USA. [4]Department of Biology, University of Ljubljana, Ljubljana, Slovenia. [5]Institute of Molecular Biology and Biotechnology, University of Crete, Heraklion, Greece. [6]Fungal Respiratory Infections Research Unit, University of Angers, Angers, France. [7]Present address: Complex Carbohydrate Research Center, University of Georgia, Athens, GA 30602, USA. [8]Present address: Department of Chemistry, University of Michigan, Ann Arbor, MI 48109, USA. ✉e-mail: rabg@uaem.mx; wangtuo1@msu.edu

*Hortaea werneckii*[13,14]. With high salinity, *W. ichthyophaga* produced three-fold thickened cell walls and bulky multicellular clumps while *H. werneckii* showed compromised cell wall integrity when melanin synthesis was inhibited[14–16].

Many *Aspergillus* species, such as *A. atacamensis, A. destruens, A. flavus, A. niger, A. tubingensis, A. versicolor*, and more recently *A. sydowii*, have been examined to understand their growth at high NaCl concentrations[17,18]. *A. sydowii* is an ascomycetous filamentous fungus found in various habitats, including salterns, dried food, decaying plant matter, and sea water, where it is a major contributor to coral disease aspergillosis[19]. Recent transcriptomic and imaging studies conducted on *A. sydowii* have demonstrated notable alternations in gene expression associated with cell wall biogenesis under high salt concentration (2.0 M NaCl) related with a thickening of the mycelial cell wall[17,20]. While these observations indirectly suggested that the remodeling of the cell wall might be crucial for fungal survival, characterizing such changes on the molecular level is challenging due to the heterogeneity and insolubility of this organelle.

Recently, the use of solid-state NMR (ssNMR) spectroscopy has led to a better understanding of the molecular architecture and dynamics of fungal cell walls[21–23]. Because intact cells are being analyzed, the structural information of the cell wall and its polysaccharides can be directly obtained at atomic resolution, without the need for solubilization or extraction[24,25]. This spectroscopic technique also provides valuable insight into the physical properties of biopolymers, including their hydrophobicity and dynamics, which provides a means to distinguish between the rigid cores and flexibility components[26,27]. Polysaccharides with high dipolar order parameters and slow relaxation are indicative of high stiff scaffolds, while biomolecules with rapid relaxation exhibit a high level of molecular motions[28–30]. Such physical profiles of biopolymers naturally complement the chemical solubility, linkage pattern, and localization of carbohydrates investigated by chemical assays and imaging techniques[31–34], which allow for a complete portrait of the cell wall organization to be assembled[21].

In the case of *Aspergillus fumigatus*, a prevalent airborne pathogenic fungus of the same *Trichocomaceae* family as *A. sydowii*, integrated ssNMR and biochemical analyses of the intact mycelia have discovered that a poorly hydrated and mechanically stiff core formed by physically colocalized chitin and α-1,3-glucan[24,35,36], which is conserved in both mycelia and conidia[37], but with altered molecular composition during morphotype transition[38]. Highly branched β-1,3/ 1,6-glucans and linear terminal threads of β-1,3/1,4-glucans comprise the mobile and well-hydrated meshes. The inner domain is shielded by a dynamic outer layer that contains galactomannan (GM), galactosaminogalactan (GAG), α-1,3-glucan, and protein components[36]. GM and GAG also covalently connect to structural proteins through linkers containing hydrophobic amino acid residues that were preserved in the alkali-insoluble fraction of the cell wall and vanished in GM- and GAG-deficient mutants[36]. These studies have reshaped our understanding in the dynamic assembly of *A. fumigatus* cell walls, shedding light onto the possible cell wall organization of the taxonomically related halophilic *Aspergillus* species such as *A. sydowii*.

Since variations in the structural organization of the fungal cell wall are often associated with alternations in environmental factors[32,39–41], such as salinity, we hypothesize that the altered cell wall organization contributes to the survival of halophiles in hypersaline conditions. Therefore, in this study, we are tailoring the high-resolution ssNMR techniques, recently developed for *A. fumigatus* cell wall characterization[35,38], to directly decipher the salinity-induced restructuring of cell walls in halophilic *Aspergillus* species. This study on moderately halophilic fungi such as *A. sydowii* also serves as the foundation for future investigations on strict halophiles such as *Wallemia*. Understanding how eukaryotic cells resist high salt levels is important in our time of global warming and water scarcity. Molecular-level insight into the modifications induced by high salt in fungi could also serve as a paradigm for other eukaryotic systems.

To achieve this goal, here we examine uniformly $^{13}$C, $^{15}$N-labeled *A. sydowii* cells (strain EXF-12860) cultured at different NaCl concentrations. The cell wall of *A. sydowii* exhibits an interlaced structure like that of *A. fumigatus* but with the addition of chitosan and the exclusion of α-1,3-glucan. These characteristics were repeatedly observed in other halophilic *Aspergillus* species examined in this study, including *Aspergillus atacamensis* and *Aspergillus destruens*. In *A. sydowii*, the amount of chitin and the proportion of amino sugars in GAG progressively increases as the salt concentration rises, with a small amount of α-1,3-glucan reintroduced to the mobile phase at the hypersaline condition. Chitosan and β-glucans are tightly associated with chitin and each other, but a high concentration of salt weakens these packing interactions and promotes the self-aggregation of biomolecules. These characteristics are repeatedly observed in other halophilic *Aspergillus* species, such as *A. atacamensis*, and *A. destruens*, which were grown under their respective optimal salt concentrations. These structural adjustments allow *A. sydowii* to produce thick and rigid cell walls with limited water permeability. The dehydration and rigidification of protein and lipid components further contribute to this effect. These molecular-level modifications in the fungal cell walls and associated organelles help the microorganisms maintain the structural integrity of their carbohydrate frame and lower water potential than their surroundings. This study elucidates the structural mechanisms employed by halophiles to withstand environmental stress and establishes a general approach for comprehending molecular-level modifications in cell walls crucial for fungal survival.

## Results

### Structural complexity of *A. sydowii* carbohydrates grown in presence of NaCl

We used *A. sydowii* as a halophile model and characterized its mycelia grown without and with NaCl at two different concentrations: optimal salinity (0.5 M) and hypersaline condition (2.0 M). The cell wall of *A. sydowii* grown at the optimal salt concentration of 0.5 M was a composite of biopolymers with distinct mobilities. We found that the rigid polysaccharides included chitin, β-1,3-glucan, and chitosan (Fig. 1a), while the mobile fraction mainly contained β-1,3-glucans, GM, and GAG (Fig. 1b). Rigid molecules were selectively detected using a two-dimensional (2D) $^{13}$C-$^{13}$C correlation spectrum that relied on dipolar-based $^1$H-$^{13}$C cross-polarization (CP) for creating the initial magnetization (Fig. 1c). The spectrum was dominated by the signals of chitin and β-1,3-glucan, such as the characteristic C1-C2 cross peak of chitin at (103.6, 55.5 ppm) and the C1-C3 cross peak of β-1,3-glucan at (103.6, 86.4 ppm). Chitosan, a deacetylated form of chitin, was also detectable, though relatively weak. These three types of polysaccharides were found to form the rigid scaffolds that share the mechanical load of the polymer network in the mycelial cell wall.

Mobile polysaccharides were detected by a combination of $^{13}$C direct polarization (DP) and a short recycle delay of 2 s in the 2D refocused *J*-INADEQUATE[42] spectrum (Fig. 1d). This technique filtered out rigid molecules with slow $^{13}$C-$T_1$ relaxation. The spectrum showed well-dispersed signals of galactopyranose (Gal*p*), galactosamine (GalN), and N-acetylgalactosamine (GalNAc), which are three monosaccharide units forming the heteroglycan GAG found on cell surfaces[43]. We also identified signals of 1,2- and 1,6-linked α-mannose (Mn$^{1,2}$ and Mn$^{1,6}$), which make up the backbone of GM, and the galactofuranose (Gal*f*) residues that form GM sidechains[44,45].

Although GM and chitin have been found to be covalently bridged through β-1,3-glucan as an integrated structural domain[31], our results identified these two molecules in two dynamically distinct fractions. This could result from the distribution of β-1,3-glucan in both rigid and mobile domains (Fig. 1c, d), where it experienced a transition from the

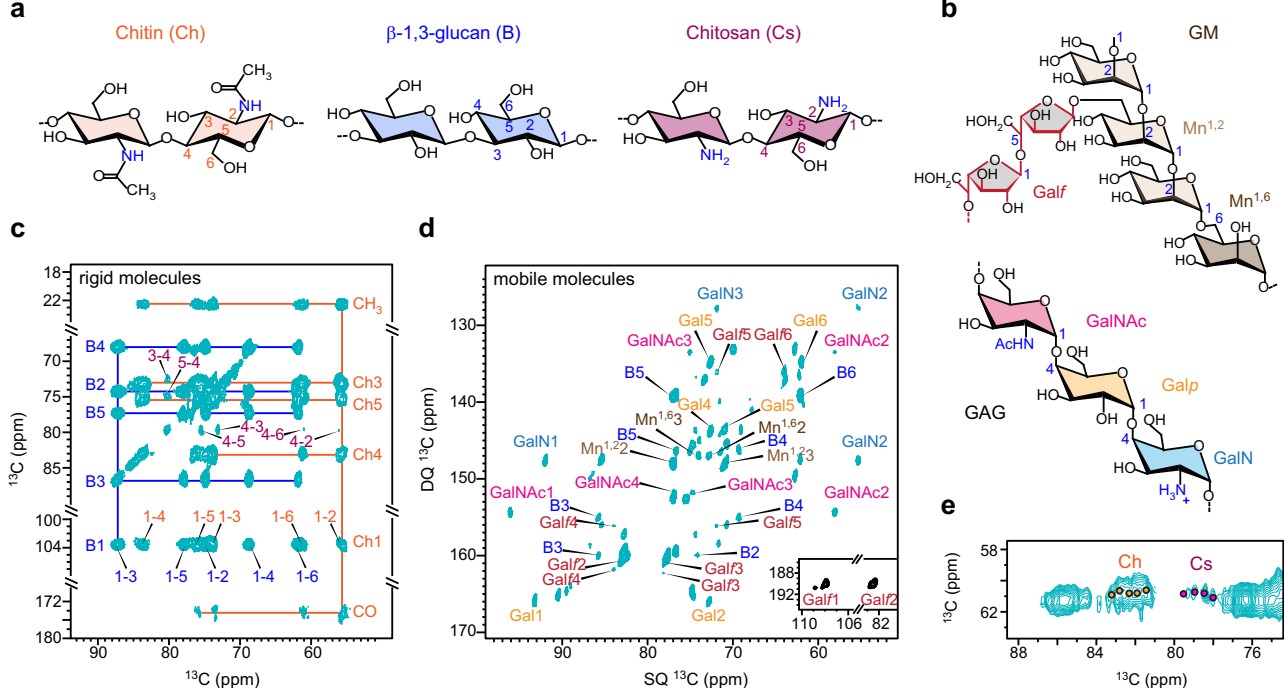

**Fig. 1 | Rigid and mobile polysaccharides of *A. sydowii*. a** Simplified structural presentation of rigid polysaccharide in the cell wall. Carbon numbers, NMR abbreviations, and color codes are given for each polysaccharide: chitin (Ch, orange), β-1,3-glucan (B, blue), chitosan (Cs, purple). **b** Representative structures of GM and GAG in the mobile domain, with key sugar units labeled: galactofuranose (Gal*f*, red), α-1,2-mannose (Mn$^{1,2}$, pale brown), α-1,6-mannose (Mn$^{1,6}$, brown), galactopyranose (Gal*p*, yellow), galactosamine (GalN, light blue), and N-acetylgalactosamine (GalNAc, magenta). **c** 2D $^{13}$C-$^{13}$C correlation spectrum of *A. sydowii* measured with CP and 100 ms DARR detecting rigid molecules. Orange and blue solid lines trace the carbon linkages of chitin and β-1,3-glucan, respectively. Each cross peak is the correlation of two carbons, such as the 1–4 cross peak in orange, which represents the correlation between carbons 1 and 4 of chitin. **d** $^{13}$C DP refocused *J*-INADEQUATE spectrum detecting mobile polysaccharides. Assignments contain NMR abbreviation and carbon number, for example, B5 represents β-1,3-glucan carbon 5. **e** Ch6-4 region of 2D $^{13}$C-$^{13}$C correlation spectrum resolving five chitin forms (orange circles) and four chitosan types (magenta circles). All spectra were measured on an 850 MHz NMR spectrometer at 13 kHz MAS on intact *A. sydowii* cells grown with 0.5 M NaCl.

rigid side that was bridged to stiff chitin to a mobile end that was connected to dynamic GM.

Polysaccharides are inherently polymeric when placed in the cellular environment. Five chitin forms and four chitosan forms were identified as clustered signals in *A. sydowii* (Fig. 1e), indicating a small range of structural variation within each molecule, probably by conformational distribution and H-bonding difference. The chemical shifts of these chitin molecules resembled those of the α-type model with antiparallel chain packing in the crystallite, while chitosan aligned with a non-flat, relaxed two-fold helix structure (called type-II chitosan)[46–48].

### Consistent cell wall composition in other halophilic *Aspergillus* species

*A. sydowii* was compared with two other halophilic *Aspergillus* species, *A. atacamensis* (strain EXF-6660) and *A. destruens* (strain EXF-10411), which were exposed to their respective optimal salt concentrations. All three halophilic species lacked the $^{13}$C-signals of α-1,3-glucans (Fig. 2a) while the five forms of chitin signals and four types of chitosan peaks were consistently identified (Fig. 2b). Therefore, the structural fingerprint of cell walls and the polymorphism of polysaccharides were unchanged across these halophilic *Aspergillus* species.

However, the cell wall composition of *A. sydowii* was found to differ from the non-halophilic fungus *A. fumigatus* (strain Ku80). The rigid polysaccharides detected in 1D $^{13}$C CP spectra showed similar spectral patterns in these two *Aspergillus* species, except for noticeable declines of peak intensities observed at 101 ppm, a signature peak of α-1,3-glucan carbon 1 (A1), and the 71–74 ppm region with mixed contributions from α-1,3-glucan and other polysaccharides (Fig. 2c).

Subtraction of the two parental spectra revealed the complete peak list of α-1,3-glucan; therefore, α-1,3-glucan was absent in the mechanical framework of *A. sydowii* cell wall. This finding was verified across different salt concentrations and through comparisons between *A. sydowii* and multiple *A. fumigatus* strains, namely Ku80, Af293, and RL578 (Supplementary Fig. 1). The observed distinctions between *A. sydowii* and *A. fumigatus* were not due to differences in culture conditions. First, the culture media used for the three *A. fumigatus* strains encompassed a broad range of differences and was cultivated under diverse growth conditions (Supplementary Table 1), and all three *A. fumigatus* strains displayed robust signals of α-1,3-glucan. Second, *A. sydowii* cultures were grown across a range of salt concentrations from 0 to 2 M, yet none of them exhibited prominent α-1,3-glucan peaks. 2D $^{13}$C-$^{13}$C correlation spectra further revealed the emergence of chitosan in the rigid core of *A. sydowii* cell wall (Fig. 2d). Chitosan should serve as a new participant in the structural scaffold of cell walls in *A. sydowii* when α-1,3-glucan is absent.

### Influence of NaCl environment on *A. sydowii* carbohydrate profile

The ultrastructure of the *A. sydowii* cell wall was examined using transmission electron microscopy (TEM) (Supplementary Fig. 2). The thickness of the cell wall was 140 nm ± 30 nm under the optimal culture condition of 0.5 M NaCl but increased to 200 ± 20 nm at the hypersaline condition (Fig. 3a). Thickening of *A. sydowii* cell wall under hypersaline conditions was consistently observed in both the current samples cultured in liquid media and in previous samples grown on wheat straw[17]. The cell wall thickness also increased upon transitioning from the optimal concentration to a salt-deprived condition, which

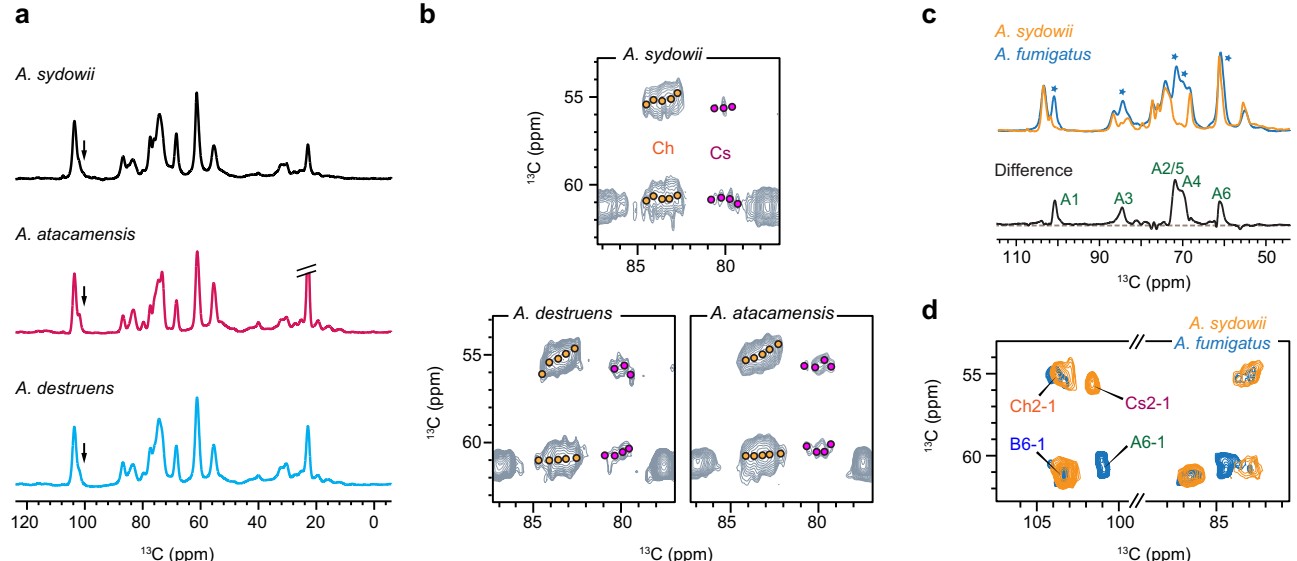

**Fig. 2 | Structural comparison of different *Aspergillus* species. a** 1D ¹³C CP spectra collected on three halophilic fungi cultured in respective optimum salt concentrations: 0.5 M for *A. sydowii*, 1.0 M for *A. atacamensis*, and 1.9 M for *A. destruens*. The expected position for α-1,3-glucan carbon 1 is marked by arrows. **b** Structural polymorphism of chitin (Ch) and chitosan (Cs). Ch2-4 and Ch6-4 cross peaks indicate five forms of chitin (orange circles) while Cs2-4 and Cs6-4 cross peaks show three to four types of chitosan (magenta circles) consistently identified in halophilic *Aspergillus* species with minor changes. **c** Rigid polysaccharides shown by 1D

¹³C CP spectra of *A. fumigatus* (cyan) cultured at 0.1 M NaCl and *A. sydowii* (yellow) cultured at 0.5 M NaCl. Asterisks indicate the positions where the peak intensities were low in *A. sydowii*. Subtraction of two parental spectra generates a different spectrum showing the α-1,3-glucan (A) signals absent in *A. sydowii*. **d** Overlay of 2D ¹³C correlation spectra collected on *A. sydowii* (yellow; 100 ms DARR) and *A. fumigatus* (cyan; 53 ms CORD). The abbreviation B is used to represent β-1,3-glucan.

stands in contrast to the observations in straw-grown samples[17]. Nonetheless, the ratio between the cell wall thickness and the total mycelial cell width steadily increased as the salt concentration in the liquid medium rises. Under osmotic stress, the stiff carbohydrate core effectively retained its structural integrity. We observed generally consistent patterns in the polysaccharide region when comparing samples cultured at varying salt concentrations, while significant differences were exhibited by proteins and lipids (Fig. 3b). This trend was consistently observed across three batches of replicates for each of the three NaCl concentrations (Supplementary Fig. 3). These batches also exhibited an identical distribution of biomolecules in dynamically distinct domains, highlighting the remarkable reproducibility of these fungal samples.

Chitin signals were initially weak in the sample lacking NaCl but became stronger in the presence of NaCl (Fig. 3c). Quantification of peak volumes revealed an upsurge in the chitin content with increasing salinity, while the amount of hydrophilic β-glucan decreased gradually (Fig. 3d and Supplementary Table 2). The introduction of more crystalline chitin to the cell wall inevitably strengthened this biomaterial.

As salt concentration increased, the amount of GM dropped substantially but the amount of GAG increased slightly (Fig. 3d, e). Surprisingly, we also observed a low amount of mobile α-1,3-glucan in the hypersaline sample, but not in optimal or salt-free conditions (Fig. 3f and Supplementary Fig. 4). Under the hypersaline condition, the contents of amino sugars, including GalNAc and GalN, were doubled compared to fungal cultures under normal and low salt conditions (Fig. 3d). The slightly acidic pH of *A. sydowii* culture was well below the GalN pKa of ~11.8; therefore, GalN should favorably occur as the cationic form GalNH₃⁺ (Fig. 1b) rather than as the conjugate base GalNH₂. The enrichment of cationic GalN units (GalNH₃⁺) in the chain should have modified the physicochemical properties of GAG and made this polymer more cationic.

## Remodeled polymer network of the cell wall
The mechanical properties and nanoscale assembly of cell walls are typically governed by the intermolecular interactions of

biomolecules[49]. Sub-nanometer polymer contacts were identified through a 2D ¹³C-¹³C correlation measured with a 1.5 s proton-driven spin diffusion (PDSD) mixing period. For example, many cross-peaks were unambiguously identified between chitin methyl groups and chitosan carbons (Fig. 4a). However, some cross-peaks observed at optimal conditions, such as the chitin carbon 4 and chitosan carbon 1 (Ch4-Cs1) and between β-1,3-glucan carbon 3 and chitosan carbon 4 (B3-Cs4) observed in Fig. 4a, disappeared in the hypersaline sample (Supplementary Fig. 5), suggesting loosened packing interfaces between chitosan and chitin/glucan at hypersaline condition.

Analysis of 30 intermolecular cross peaks uncovered the organization pattern of the polysaccharide network (Supplementary Table 3). The interactions between different carbon 4 sites of chitin units revealed the coexistence of these sub-forms in the same chitin crystallite (Fig. 4b). This feature was consistently found in both 0.5 M and 2.0 M *A. sydowii* samples. Crystalline chitin is physically supported by the β-glucan matrix and can also covalently link to β-glucan and then to GM, as reported by NMR and chemical assays of *A. fumigatus*[31,36]. Although the semi-dynamic β-glucan was disfavored in long-range correlation experiments, its carbon 3 and carbon 5 still showed strong cross peaks with the carbon 5 and methyl of chitin, regardless of the salt concentration. Under optimum salt concentration, chitosan was mixed with both chitin and β-glucan, but such contacts became limited in the hypersaline habitat. The hyperosmotic condition induced the restructuring of fungal cell walls.

## Changes in water accessibility and polymer dynamics
The fungal cell wall has dramatically modified its water accessibility and polysaccharide dynamics in response to varying salt concentrations. Water accessibility refers to the number of immobilized water molecules present at each carbon site, while polysaccharide dynamics pertain to the movement of these molecules on the nanosecond and microsecond timescales. Polymer hydration was investigated in a site-specific manner using a 2D ¹³C-¹³C correlation water-edited experiment[50,51] that selectively detected the signals of water-

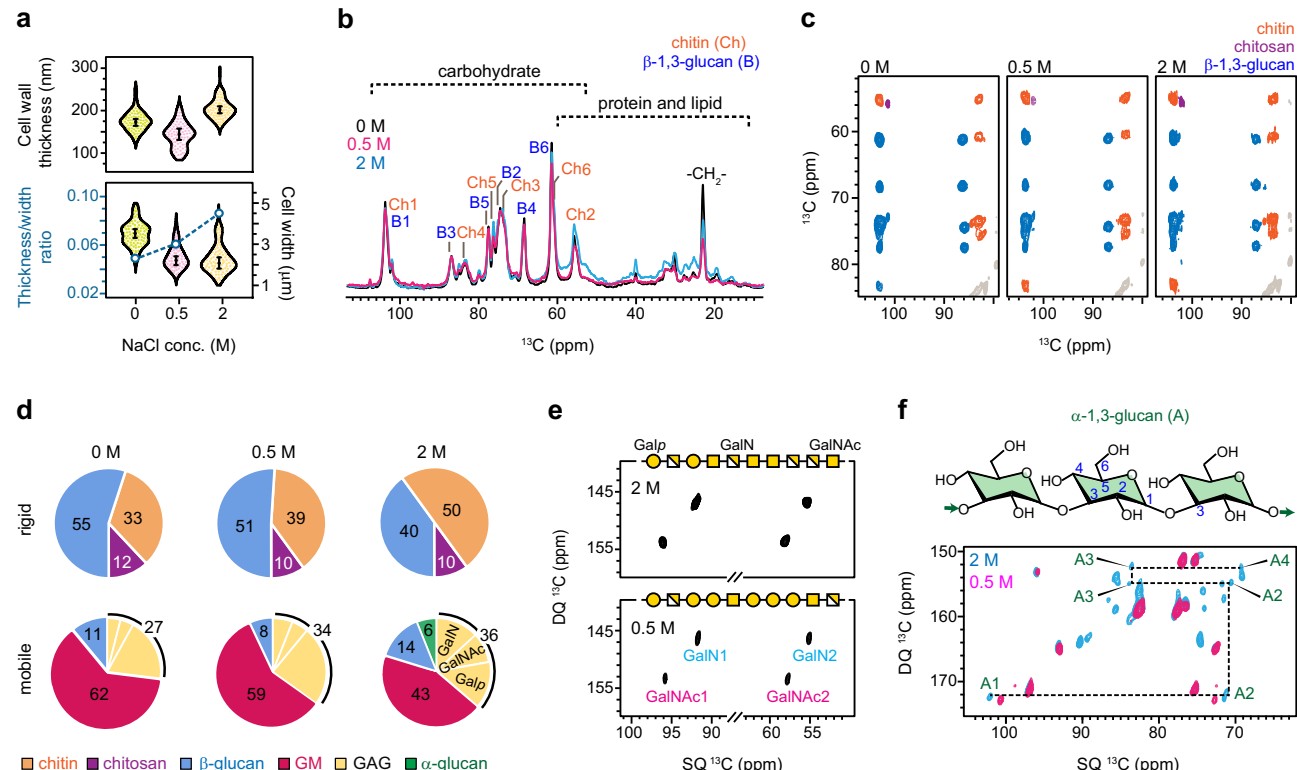

**Fig. 3 | Effect of salt concentration on *A. sydowii* polysaccharide composition.**
**a** Distribution of cell wall thickness (top panel) and its relative ratio to the cell thickness (bottom panel) in *A. sydowii* hyphae exposed to different NaCl concentrations. Each violin plot of cell wall thickness depicts 100 measurements from 10 cells ($n = 100$), with the average value and error bars (standard deviation) presented. The statistical significant differences ($\alpha = 0.05$) were identified by unpaired Student $t$ test. The ratios of cell wall thickness to cell width were shown using blue open circles and connected by dash lines (left axis) while the violin plots of cell width values are projected to the right axis. $n = 100$ (10 cells) for either the 0 M or 2.0 M sample and $n = 70$ (7 cells) for the 0.5 M sample. **b** Comparison of 1D [13]C CP spectra of *A. sydowii* cultures at 0 M, 0.5 M, and 2.0 M NaCl. Key features of carbohydrate and protein/lipid signals are labeled for chitin (Ch), β-1,3-glucan (B), and the CH2 of lipid acyl chain. **c** 2D [13]C-[13]C DARR correlation spectra of *A. sydowii*

samples, with chitin signals (orange), β-1,3-glucan signals (blue), and chitosan (purple) signals marked. The relative abundance of chitin increases at high salt concentrations. **d** Molar composition of the rigid (top row) and mobile (bottom row) polysaccharides in *A. sydowii* cell walls, determined by peak volumes of 2D [13]C CP DARR and [13]C DP *J*-INADEQUATE spectra, respectively. The fractions of Gal*p*, GalN, and GalNAc in GAG are also shown. **e** Stronger signals of GalN and GalNAc units in GAG at the higher salt concentration in [13]C DP *J*-INADEQUATE spectra. GAG structures are constructed following the molar fraction using the Symbol Nomenclature for Glycans. **f** Structure of α-1,3-glucan (A) and carbon connectivity tracked by [13]C DP *J*-INADEQUATE spectra. α-1,3-glucan is barely detectable in 0.5 M NaCl condition but becomes visible in 2.0 M NaCl condition. Source data of Fig. 3a, d are provided as a Source Data file.

associated biomolecules (Supplementary Fig. 6). The intensity of the water-edited signals (S) was compared to the equilibrium condition ($S_0$) to determine the $S/S_0$ ratio for each carbon site, which is an indicator of water retention (Supplementary Table 4).

The normalized $S/S_0$ ratios were substantially higher for β-glucan than for chitin within each *A. sydowii* sample (Fig. 4c). This observation revealed the different structural roles of these polysaccharides: chitin constitutes the hydrophobic center, while β-glucans form the hydrated matrix. The data revealed that *A. sydowii* cell walls were best hydrated at the optimal concentration of 0.5 M NaCl. Specifically, the average $S/S_0$ ratios for β-glucans and chitin are 0.51 for and 0.20. respectively. However, the extent of water association dropped substantially at 0 M and 2.0 M NaCl concentrations (Fig. 4c), both of which are considered stress conditions for *A. sydowii*[20]. In the absence of NaCl, the hydration level of chitin remained unchanged but the $S/S_0$ ratio of β-glucan dropped by more than one-third. Under hypersaline conditions, both chitin and β-glucan were poorly hydrated, with $S/S_0$ ratios of 0.18 and 0.39, respectively.

The motional characteristics of cell wall polysaccharides were determined using NMR relaxation experiments (Supplementary Fig. 7 and Table 5). A molecule with fast [13]C-$T_1$ relaxation is highly dynamic on the nanosecond (ns) timescale, likely due to rapid local reorientation motions (Fig. 4d). Similarly, molecules exhibiting fast [1]H-$T_{1\rho}$ relaxation

are mobile on the microsecond (μs) timescale, typically attributed to slower collective movements and flipping (Fig. 4e). Within each sample, β-glucan showed shorter [13]C-$T_1$ and [1]H-$T_{1\rho}$ time constants than chitin, demonstrating the dynamic nature of β-glucans.

When we deviated from the optimal condition of 0.5 M to either 0 M or 2.0 M, both chitin and β-glucans showed longer [13]C-$T_1$ (Fig. 4d) and shorter [1]H-$T_{1\rho}$ (Fig. 4e). The average [13]C-$T_1$ increased from 1.6 s to 1.8–2.0 s for chitin and increased from 1.0 s to ~1.2 s for β-glucan. Meanwhile, the average [1]H-$T_{1\rho}$ dropped from 14 ms to 10–12 ms for chitin and from 12 ms to 9–10 ms for β-glucan, likely caused by the loosened interface between different polymers. Therefore, in salt-free or hypersaline environments, biopolymers in the inner cell wall have restricted reorientation motions on the nanosecond timescale but accommodate slower and larger-scale movements on the microsecond timescale. Even though the centesimal composition of the cell wall polymers was different at 0 and 2 M NaCl, the biophysical data showed that polymer dynamics and hydration, as well as cell wall thickness, lead to similar changes in the cell wall assembly when deviating away from the optimal concentration.

## Protein and lipid components
We observed strong signals from proteins and lipids, which could have originated from various sources, including cell walls and plasma

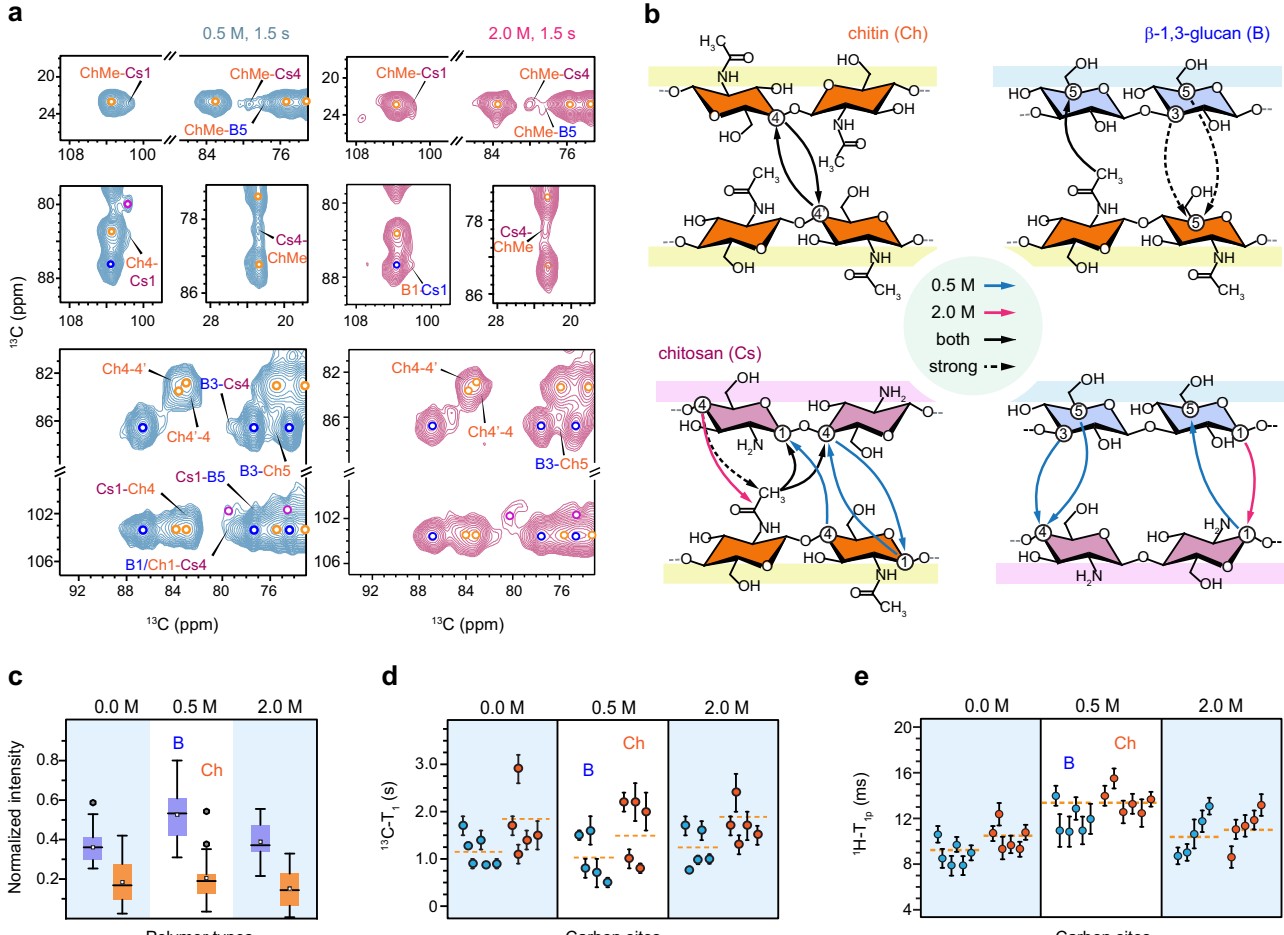

**Fig. 4 | Packing, hydration, and dynamics of *A. sydowii* polysaccharides.**
**a** Intermolecular cross peaks identified in 2D $^{13}$C correlation spectra measured with long (1.5 s PDSD) mixing periods on 0.5 M (left) and 2.0 M (right) samples. Signals of chitin (Ch; orange), β-glucan (B; blue) and chitosan (Cs; purple) are marked by open circles. Intermolecular peaks are labeled. **b** Summary of intermolecular cross peaks observed in *A. sydowii*. Arrows show the direction of polarization transfer. Blue and magenta lines show the interactions observed only in 0.5 M and 2.0 M conditions, respectively. Black solid lines and dash lines represent interactions observed in both samples in both 1.5 s and 0.1 s PDSD spectra, respectively. **c** Box-and-whisker diagram plotting the relative intensities (S/S₀) of β-1,3-glucan (blue; $n$ = 24, 25, 25) and chitin (orange; $n$ = 17, 15, 14) in three *A. sydowii* samples with varying salt

concentrations. The box displays the interquartile range (25th to 75th percentiles) of the dataset. Whiskers show the 5th and 95th percentiles, marking outliers as open circles. The central line denotes the median (50th percentile) and the open box represents the mean. **d** $^{13}$C-$T_1$ relaxation time constants of β-1,3-glucan (blue) and chitin (orange) in *A. sydowii*. The average $^{13}$C-$T_1$ are marked using yellow dash lines. **e** $^1$H-$T_{1\rho}$ relaxation times of β-1,3-glucan (blue) and chitin (orange). The average values over all carbon sites within a polysaccharide are shown by dash lines. For panels d and e, error bars indicate standard deviations of the fit parameters of $^{13}$C-$T_1$ ($n$ = 10) and $^1$H-$T_{1\rho}$ ($n$ = 12) relaxation times respectively. **c–e**, the dataset of 0.0 M and 2.0 samples are shaded in blue for better comparison with the 0.5 M optimal condition. Source data of Fig. 4c–e are provided as a Source Data file.

membrane components, as well as intracellular organelles. We found that the protein and lipid components mainly reside in the mobile phase (Supplementary Fig. 8). The signals of amino acids were distinguished using 2D refocused *J*-INADEQUATE spectra (Fig. 5a). As protein backbone chemical shifts are sensitive to $\varphi$ and $\psi$ torsion angles[52], we determined the secondary structure by comparing the observed Cα chemical shifts to random-coil values. We found that mobile proteins were predominantly in α-helical conformation, which remained consistent across the salt gradient (Fig. 5b).

By exclusively selecting rigid molecules in structurally robust components, we noticed a distinctive and plentiful presence of proteins and lipids at 2 M NaCl condition (Fig. 3b and Supplementary Fig. 9). The amino acid residues identified in this inflexible portion had a noticeable contribution to the β-strand conformation and experienced substantial dehydration in hypersaline condition (Fig. 5c and Supplementary Table 6). The rigidification and dehydration of both protein and lipid components have suggested a global change to the cell wall and its adjacent layers, including the underlying membranes and the surface hydrophobins. These spectroscopic results also

support the hypothesis that halophilic fungi differentiate the expression of hydrophobin genes to moderate surface tension and water penetration[17,53,54].

The lipid components were also examined using the 2D $^1$H-$^{13}$C refocused Insensitive Nuclei Enhanced by Polarization Transfer (INEPT) experiment (Fig. 5d)[55]. Spectral superposition of *A. sydowii* lipids and model compounds in the glycerol/headgroup region confirmed the presence of phosphatidylcholines (PC) and phosphatidylglycerols (PG) (Supplementary Fig. 10). Sterols and polyisoprenoids were not detectable in either mobile or rigid portion, likely due to their relatively low abundance in a cellular sample. We spotted the putative signals of triglyceride (TG), which became pronounced in 0 M and 2.0 M NaCl conditions (Supplementary Fig. 11 and Table 7). This molecule has been identified in multiple *Aspergillus* and *Cryptococcus* species and was reported to modulate membrane fluidity[38,56]. However, due to the severe overlap of its putative signals with those from other lipids and proteins, and due to the broad distribution of lipid polymers in the cell, more biochemical studies should be undertaken here to explain these changes.

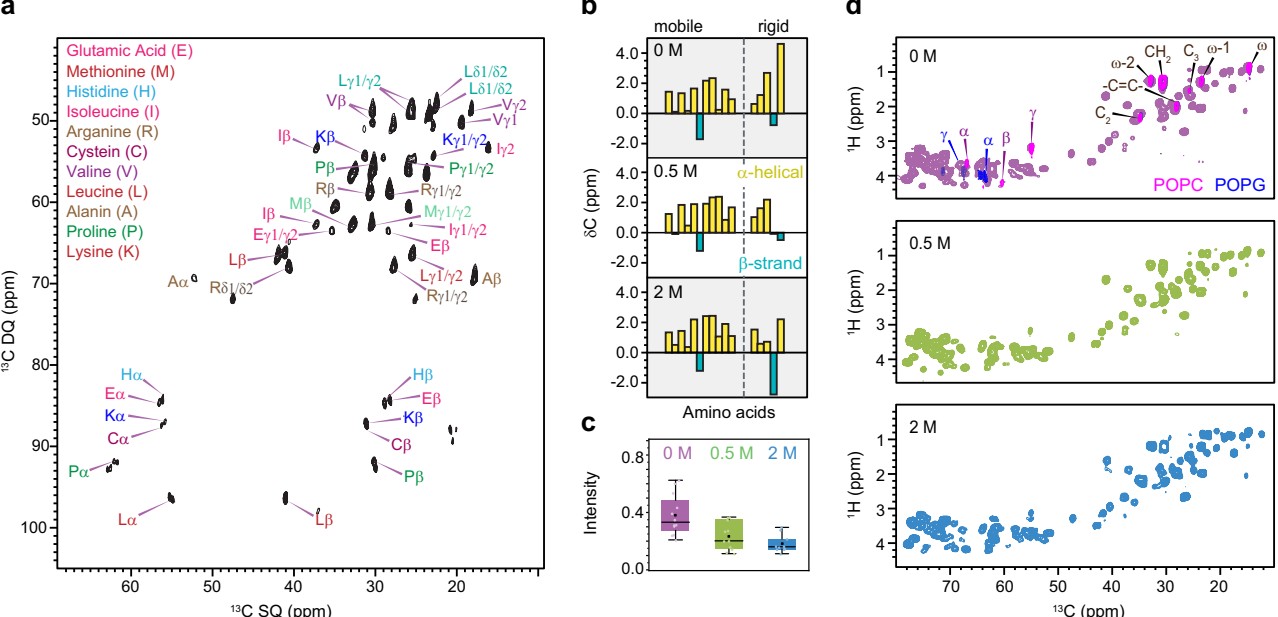

**Fig. 5 | Fingerprints of *A. sydowii* proteins and lipids. a** Protein region of DP refocused J-INADEQUATE spectra collected using *A. sydowii* (0.5 M NaCl). **b** Secondary structure of proteins denoted by $^{13}$C chemical shifts of Cα. α-helical and β-strand conformations are in yellow and blue, respectively. The amino acid residues in the mobile fraction (left) and rigid fraction (right) are separated by dash lines. **c** Box-and-whisker diagrams plotting of the relative water-edited intensities (S/S$_0$) of protein carbon sites in 0 M (purple; $n = 11$), 0.5 M (green; $n = 12$) and 2.0 M (blue; $n = 10$) in three *A. sydowii* samples with varying salt concentrations. The box contains the 25th to 75th percentiles of dataset. The black center line denotes the median value (50th percentile) and the open box represents the mean. The whiskers mark the 5th and 95th percentiles, with values beyond these upper and lower bounds considered outliers (open circles). All the data points are shown in each concentration. **d** 2D refocused INEPT $^1$H-$^{13}$C correlation spectra of *A. sydowii* samples cultured with 0 M, 0.5 M, and 2.0 M NaCl. The spectra are compared with the control spectra of model lipids POPC (magenta) and POPG (blue), showing the α, β, and γ carbons in phospholipid headgroups and the carbons in lipid tails. Source data of Fig. 5c are provided as a Source Data file.

## Discussion

In this study, we conducted high-resolution ssNMR analysis to unveil the molecular-level organization of *A. sydowii* cell walls, which has been summarized in Fig. 6a. At optimum salt concentration, the inner cell wall of *A. sydowii* was found to contain rigid chitin and chitosan in partially crystalline and highly polymorphic structures[47], surrounded by a matrix mainly consisting of β-glucans that regulate the water accessibility of the cell wall mesh in the absence of α-1,3-glucan. Chitin and β-glucan, along with chitosan, are well mixed on the nanoscale, with extensive intermolecular interactions as shown by long-range correlation data (Fig. 4b). This inner domain is covered by an outer shell rich in highly dynamic molecules, mainly containing GM and GAG. Previous chemical assays of *Aspergillus* cell walls showed a carbohydrate core formed by covalently linked chitin-β-glucan-GM complex[31,33], which could explain the NMR-observed bimodal distribution of β-1,3-glucan in both rigid and mobile domains (Fig. 3d). The rigid segment is in contact with chitin or chitosan, while the mobile part forms the soft matrix and bridges to even more dynamic GM in the outer shell (Fig. 6a).

The key discovery is the direct observation of the molecular-level changes in the cell wall structure when exposed to hypersaline conditions (Fig. 6b). Our experimental data clearly demonstrate that the fungus has developed a thickened, stiff, waterproof, and adhesive cell wall for better survival in hypersaline habitats with restricted water activity[18,19]. The inner domain of the cell wall contains more chitin molecules, which provide high rigidity, and less β-1,3-glucans, which abolish water permeability. Analyses of intermolecular interactions (Fig. 4b) have shown that the packing interactions between chitin-chitin and chitin-glucan remain unchanged. However, chitosan becomes better isolated from other molecules, possibly due to self-aggregation. The surface layer has a reduced amount of GM but an increased content of α-1,3-glucan and GAG with an enriched fraction of cationic GalN (GalNH$_3^+$). This chemical change is crucial for facilitating its adherence to anionic surfaces, including human cells, and promoting the adhesion between mycelia, which helps the entire colony withstand unfavorable conditions[43,57–59].

Our data has shown a wide distribution in both rigid and mobile fractions of the *A. sydowii* cell wall for the biopolymers contributing to the formation of the GM-β-1,3-glucan-chitin complex (Fig. 1d). It is also possible that this covalently linked complex has a relatively low abundance, and therefore, the observed polymer dynamics may be predominantly governed by the individual polysaccharides that exist separately within the bulk of the cell wall. Previous chemical results have demonstrated that the GM-β-1,3-glucan-chitin complex constitutes only approximately 7% of the total cell wall weight in *A. fumigatus*[60]. Because *A. sydowii* and *A. fumigatus* are taxonomically close, we can expect a similarly low amount of this polysaccharide complex in the cell wall of *A. sydowii*.

Previous studies on *A. sydowii* showed differential transcriptional expression of many genes under environmental perturbations[17,18,20,61]. However, when specifically analyzing the expression of the orthologs of *A. fumigatus* genes known to be involved in cell wall polysaccharide synthesis[62], it became challenging to reconcile these results with the structural alternations in *A. sydowii* cell walls revealed by ssNMR analysis. The transcriptomic data only correlates with our NMR-detected changes in chitin and β-1,3-glucan synthesis induced by the high salt concentration, and cannot explain the variations in the content of α-1,3-glucan, GAG, or GM observed in this study.

We anticipated a strong correlation between the transcriptomic data and the evolution in α-1,3-glucan and GAG concentrations when exposed to high salt levels, because these two polysaccharides are found in the outer layer of the cell wall and play a pivotal role in regulating the permeability of molecules in *A. fumigatus*[57,63]. The NMR-observed increases in GAG and α-1,3-glucans agreed with the decrease

**a**

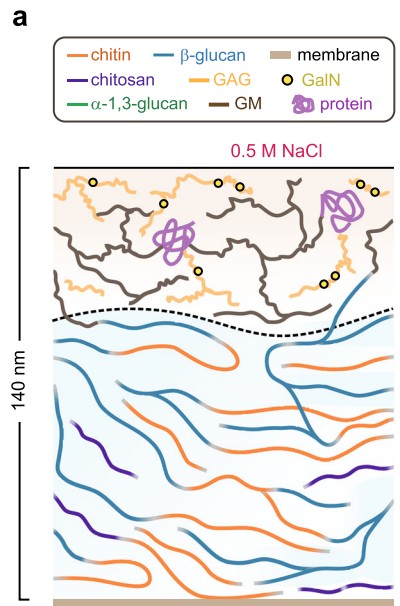

**b**

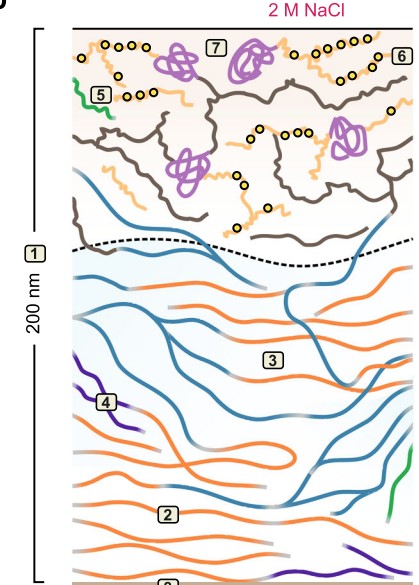

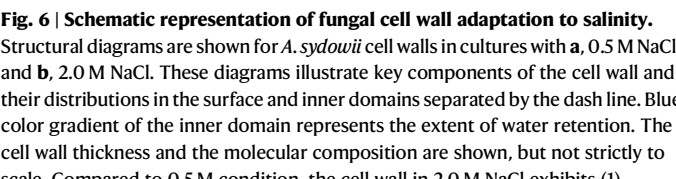

**Fig. 6 | Schematic representation of fungal cell wall adaptation to salinity.**
Structural diagrams are shown for *A. sydowii* cell walls in cultures with **a**, 0.5 M NaCl, and **b**, 2.0 M NaCl. These diagrams illustrate key components of the cell wall and their distributions in the surface and inner domains separated by the dash line. Blue color gradient of the inner domain represents the extent of water retention. The cell wall thickness and the molecular composition are shown, but not strictly to scale. Compared to 0.5 M condition, the cell wall in 2.0 M NaCl exhibits (1)

increased thickness, (2) enhanced biosynthesis of crystalline chitin resulting in higher cell wall rigidity and restricted local motions in the inner domain, (3) reduced water retention due to lower β-glucan content, (4) chitosan aggregation and reduced interactions with other components, (5) inclusion of α-glucan in the mobile phase, (6) enriched content of cationic GalN units in GAG on the surface, (7) increased protein content and rigidity, dehydration of protein, and reshuffled secondary structure, and (8) elevated content of rigid lipids.

in *A. sydowii* cell wall permeability to water in a 2.0 M NaCl environment. However, all orthologs of α-1,3-glucan synthase genes controlling α-1,3-glucan synthesis were expressed across various salt concentrations, as shown in the previous study[62], even though α-1,3-glucan was only detected in *A. sydowii* cell walls in the presence of 2 M NaCl in this study. Additionally, the expression levels of the orthologs of endo-α-1,3-glucanase genes remained unaffected by salt concentrations. Similarly, the expression of the genes within the GAG synthesis cluster was rather downregulated as shown before ref. 62, despite a notable increase in the amount of GAG and especially the levels of GalN and GalNAc as observed here under hypersaline conditions.

The NMR-detected decline in GM content cannot be explained by a decrease in the expression of UDP-Gal mutase or the genes regulating the elongation of the galactofuran chains, because all the orthologs of these genes from the *A. fumigatus* genome were upregulated at hypersaline conditions in the previous study[62]. Furthermore, the orthologs of the DFG family, which play a critical role in incorporating GM into the β-1,3-glucan-chitin core, were not significantly downregulated[64,65].

The observed lower content of β-1,3-glucan in the presence of salt cannot be explained either by a reduced expression of the orthologs of the β-1,3-glucan synthases nor the UTP-glucose-1-phosphate uridylyltransferase that is responsible for the production of UDP-glucose, the substrate of the β-1,3-glucan synthase. However, expression of the ortholog of β-1,3-glucan elongases was significantly downregulated as reported before[62]. This result confirmed that the transglycosidases from the Gel family play a major role in the elongation of β-1,3-glucans after oligo-β-1,3-glucans have been synthesized by the glucan synthase complex and extruded from the plasma membrane[62,66,67]. This would reinforce the role of this GPI-anchored protein family in cell wall construction. While many exo and endo β-1,3-glucanases genes were identified, their level of expression was very variable, and it remained impossible to discriminate between the glucanases involved in the degradation of

extracellular substrates and the endolysin proteins that may putatively degrade the cell wall glucans.

The salt-induced increase in chitin content reported here was not linked to an upregulation of the chitin synthases orthologs of *A fumigatus*[68]. However, the expression of the genes governing UDP-GlcNAc production[69] were significantly stimulated when *A. sydowii* was grown in hypersaline condition. This suggests a need to investigate the transcriptional regulation of the UDP-GlcNAc synthesis during fungal stress. Not all the chitinase genes were downregulated, which led to the question of the putative endolytic activity of only some of these chitinases while the other would be only associated to the exochitinase catabolic activity[70,71]. This result confirms the limited understanding of the role of glycosyl hydrolases during cell wall construction.

Investigating the roles of transcription factors and regulators in the cell wall synthesis of *A. sydowii* under hypersaline condition remains incomplete. Strikingly, none of the MAP kinase HOG orthologs, known to regulate the pathways of fungal adaptation to salinity, were differentially expressed in response to salt in *A. sydowii*[20]. Other important pathways that govern cell wall biosynthesis, including the Ca²⁺/calcineurin pathway, the protein kinase C pathway, and the pH sensing RIM101 pathway, have not yet been explored in *A. sydowii*. Transcript profiling experiments in yeasts subjected to cell wall perturbing agents have identified a core set of regulatory genes, whose orthologs should be further analyzed in this filamentous fungus[71].

The discrepancies between ssNMR data and transcriptome raised numerous questions about cell wall synthesis and prompted hypotheses for cell wall pathways not yet discussed in the field. For example, are there any unknown inhibitory metabolites, post-translational modifications, or essential partners of the protein complexes controlling the synthesis of α-1,3-glucan synthesis, GAG, and GM? Comparison of transcriptome and cell wall composition in the case of chitin also underscores the significance of UDP-GlcNAc synthesis pathway in cell wall construction and emphasizes the need for further investigation into the pathways providing substrates to polysaccharide synthases.

The halophilic *Aspergillus* species investigated in this study exhibit distinctive characteristics when compared to the extensively studied model fungus *A. fumigatus*. One of these notable differences is the previously underemphasized structural role of chitosan, primarily due to its low abundance in *A. fumigatus*. However, the chitosan in *A. sydowii* has gained significance in cell wall organization by interacting with both chitin and β-glucans, as supported by the strong intermolecular cross peaks among these polymers (Fig. 4b), and such packing interactions has been weakened in the hypersaline condition. The unique occurrence of chitosan in *A. sydowii* in absence or stimulated in presence of salt may play a major role in halophily. Indeed, the septuple deacetylase mutant of *A. fumigatus* did not show any increase in the resistance to elevated salt (up to 0.8 M NaCl) or sorbitol concentrations (up to 1.6 M) compared to the parental strain[72].

The deficiency of α-1,3-glucan in *A. sydowii*, *A. atacamensis*, and *A. destruens* (Fig. 2a) represents another significant departure from our previous understanding based on *A. fumigatus*[36]. This can be correlated with the low transcription of α-1,3-glucan synthase genes in *A. sydowii*, as reported recently[17], or it could be due to physiological responses that hinder α-1,3-glucan biogenesis. In *A. fumigatus*, α-1,3-glucan serves as a versatile building block distributed in both the rigid and mobile phases of both the alkaline soluble and insoluble fractions[36]. α-1,3-glucans thus supports mechanical properties of *A. fumigatus* cell walls by interacting with chitin and enhances fungal virulence by concealing the β-glucans to impede immune recognition[21,34,73]. Therefore, the absence of α-glucan in halophilic species could explain the moderate virulence of these fungal strains in pathogenicity[74]. For instance, *A. sydowii* is primarily recognized as a pathogen in coral reefs, while *A. destruens* is mainly known to be an opportunistic animal pathogen[75]. In addition, the deficiency of α-1,3-glucan observed in halophilic species necessitates other molecules, such as β-glucans, to play a more prominent role in stabilizing the cell wall assembly.

In *A. sydowii* mycelia obtained under hypersaline conditions, proteins, and lipids also become rigidified and dehydrated (Figs. 2b and 4c), similar to the observed changes in cell wall polysaccharides. These effects may occur to the two layers of hydrophobins and cell membrane that sandwich the cell wall, as well as the protein and lipid components included in the macromolecular assembly of the cell wall itself. This finding echoes the earlier report that the upregulation of hydrophobin genes in *A. sydowii* samples cultured in both 2.0 M and 0 M NaCl concentrations[15,17]. Hydrophobins, with a precise sequential pattern of eight cysteine residues forming four disulfide bonds, adopts an amphipathic tertiary structure[76]. This allows hydrophobin to self-assemble into amphipathic layers to promote cells adhesion to hydrophobic surfaces, regulate solute movement, and enhance cell wall rigidity. Such properties aid the cell in withstanding mechanical strain resulting from the functions in the surrounding osmolarity[15]. The protein dehydration and stiffening observed through NMR play a role in limiting cell wall permeability and protecting the organism from the stressful environment, contributing to the adaptation of *A. sydowii* as a successful halophile.

It is notable that the limited water permeability, and altered motional characteristics were consistently observed in both 0 M and 2.0 M NaCl conditions (Fig. 4c–e), revealing a general mechanism of cell wall restructuring to resist external stress. The observed non-directional variations cannot be easily correlated with the sequential changes in the polymer composition. The cell wall of *A. sydowii* grown in high salinity became more hydrophobic, which helps to prevent water loss from the cytoplasm. This is likely due to a lower content of β-glucans, as shown in Fig. 3d. However, the NaCl-free sample with a β-glucan-rich cell wall still exhibited limited exposure to water. This observation is intriguing and may be related to the increased thickness of the cell wall (Fig. 3a), which suggests a change in the molecular assembly of the cell wall or other associated constituents. This is a paradigm in cell wall biology where similar cell wall modifications only indicate the presence of stress regardless of the nature of the stress encountered by the fungus.

These molecular-level insights unveiled the structural mechanisms employed by halophiles to cope with osmotic stress. This has implications for the application of these microorganisms in agricultural and biotechnological applications under hypersaline environments that are unfavorable for microbial growth[8,9]. Halophilic fungi have shown great potential in converting agricultural waste to fermentable sugars and remediating hypersaline soils and improving salt-related damage[6,77]. The identified structural adaptations, including the augmentation of surface charge and the elevation of rigid and hydrophobic molecules, offer valuable targets for the rational engineering of fungal strains to optimize their capability for survival through these applications, or for the development of improved solutions against aspergillosis for preserving coral ecosystems[78]. The structural features identified in *A. sydowii* can also be used to select promising microbial candidates for supporting human expeditions to extreme environments[9].

## Methods

### Culture conditions of *A. sydowii*

*A. sydowii* strain EXF-12860 was used as the primary model fungus in this study. It was isolated from solid fermentation of sugarcane bagasse[6] and was obtained from the EX microbial Culture Collection of infrastructural Center Mycosmo (MRIC UL) at the University of Ljubljana (Slovenia). The fungal strain was routinely propagated and preserved in Potato Dextrose Agar (Catalogue # CM0139B, Thermo Fisher Scientific) supplemented with 0.5 M NaCl (optimum concentration) for seven days at 28 °C. For isotopic labeling, *A. sydowii* was grown in 100 mL of liquid media containing 20 g/L $^{13}$C-glucose (Catalogue # CLM-1396-PK, Cambridge Isotope Laboratories) and 2 g/L $^{15}$N-labeled NH$_4$NO$_3$ (Catalogue # 366528, Millipore Sigma) as the only labeled carbon and nitrogen sources, together with other salt and trace elements (Thermo Fisher Scientific) as detailed in Supplementary Table 8. Approximately the same sized (~50 mm$^2$) agar plugs (diced agar pieces) were inoculated into the autoclaved liquid culture media. The culture was grown at 28 °C with 150 rpm shaking for seven days in a shaking incubator (Product # 6753, Corning, LSE). 150 rpm was chosen to prevent the excessive production of spores and viscous culture rheology. The fungus was grown in parallel using 0.5 M NaCl (optimal conditions) and under two stress conditions of 0 M (hypoosmotic) and 2.0 M (hyperosmotic) NaCl, as reported previously[17,20]. The mycelium was collected and washed twice with deionized water, and later washed with PBS (Catalogue # J62692, Thermo Fisher Scientific) to remove the excess isotope-labeled molecules and NaCl. The harvested fungal mycelia were used for both ssNMR and TEM experiments. To ensure reproducibility, three separate batches of samples were prepared for each of the three NaCl concentrations, resulting in nine $^{13}$C,$^{15}$N-labeled samples for *A. sydowii*. The NMR fingerprints of all the samples exhibited a high level of reproducibility across different batches of samples, and across different dynamic gradients within each sample (Supplementary Fig. 3).

### TEM imaging of cell wall thickness and morphology

The *A. sydowii* mycelia obtained from the three culture conditions used in this study (0 M, 0.5 M, and 2.0 M NaCl) were prepared for TEM imaging. The fungal samples were fixed for 12 h at 4 °C using 2.5% glutaraldehyde and 2 % paraformaldehyde (Catalogue # 15700, Electron Microscopy Sciences, Hartfield, PA) in 0.1 M phosphate buffer (pH 7.4) to halt metabolic processes and preserve the cells. The mycelia were then embedded in 3% agarose gel (Catalogue # 9012-36-6, Millipore Sigma) and rinsed four times with 0.1 M phosphate buffer pH 7.4 and 0.05 M glycine (Catalogue # 56-40-6; Millipore Sigma). After washing, the samples were fixed with 2% OsO$_4$ (SKU 19152, Electron

Microscopy Sciences, Hartfield, PA) in the dark for 1 h and rinsed three times using deionized water. En Bloc staining in 1% uranyl acetate (Catalogue # 22400, Electron Microscopy Sciences, Hartfield, PA) was used to increase the contrast. Dehydration was achieved using 70% ethanol series and propylene oxide for two times followed by infiltration in propylene oxide:Epon resin series. Ultra-thin sections for TEM were cut on a Dupont Sorvall MT-2 microtome. TEM sections were mounted on carbon-coated copper grids (EMS FCF-150-CU) and stained with 2% uranyl acetate and Reynolds lead citrate (Catalogue #17800, Electron Microscopy Sciences, Hartfield, PA). Measurements were performed on the perpendicular cross-sections of 100 hyphae per culture condition using a JEOL JEM-1400 electron microscope (Peabody, MA) with an accelerating voltage of 120 kV at varying magnification and photographed with 100 with Gatan Orius 1000 A camera. TEM imaging was performed at the Shared Instrumentation Facility at Louisiana State University). The TEM images were viewed, and thickness was measured using ImageJ V1.8.0_172.

Ten cells were used for measurements taking 10 cell wall thickness measurements from each cell. The statistical unpaired two-tailed student $t$ test ($p < 0.05$) was performed to compare the cell wall thickness between two concentrations (0–0.5 M, 0.5–2.0 M, 0–2.0 M). Statistical analysis was done using Microsoft Excel 365 and violin plots were generated using Origin Pro 2019b software.

## Solid-state NMR analysis of *A. sydowii* carbohydrates and proteins

SsNMR experiments were conducted using experimental schemes applied to fungal cell walls in multiple recent studies[22,35,36,38,56]. Methods include $^{13}$C-$^{13}$C through-space correlations using PDSD, DARR, and CORD sequences[79], and through-bond correlation experiments such as refocused *J*-INADEQUATE and refocused INEPT[42,55]. These methods can efficiently provide information on the polymorphic structure, composition, and physical packing of biomolecules in native cell walls, but are also limited by the long experimental time required for finishing a complete analysis. Recent development of proton-detection methods and sensitivity-enhancing dynamic nuclear polarization (DNP) techniques could expedite future analysis of fungal cell walls[23,37,80].

For ssNMR analysis of *A. sydowii*, 30 mg and 100 mg of mycelia were packed into 3.2 mm and 4 mm MAS rotors, respectively. All 1D experiments on the three replicates on an 800 MHz (18.8 Tesla) Bruker Avance Neo spectrometer at 13 kHz MAS at 298 K (Supplementary Fig. 3). 1D and 2D solid-state NMR experiments were performed on one sample per salt concentration on a Varian VNMRS 850 MHz (19.9 Tesla) spectrometer using a 3.2 mm MAS triple-resonance HCN probe under 13 kHz MAS at 290 K. The spectra were collected in Topspin 3.5 on the 800 MHz Bruker Avance Neo spectrometer and in OpenVNMRJ 2.1a on the Varian VNMRS 850 MHz spectrometer. Analysis and processing were done in Topspin 4.0.8. Water-editing, relaxation, and $^1$H-$^{13}$C refocused INEPT experiments were conducted on a Bruker Avance 400 MHz (9.4 Tesla) spectrometer under 10 kHz MAS at 293 K. The $^{13}$C chemical shifts were externally referenced to the adamantane $CH_2$ signal at 38.48 ppm on the tetramethylsilane (TMS) scale. The typical radiofrequency field strengths were 83 kHz for $^1$H hard pulses and decoupling, and 50–62.5 kHz for $^{13}$C pulses, unless otherwise specified. The key experimental parameters are listed in Supplementary Table 9.

The initial magnetization for the experiments was created in three ways: (1) using $^1$H-$^{13}$C cross-polarization to preferentially detect rigid molecules, (2) using $^1$H-$^{13}$C refocused INEPT to select the most mobile molecules[55], and (3) using $^{13}$C direct polarization to selectively detect mobile molecules with a short recycle delay of 2 s, or to quantitatively probe all carbons and molecules with a long recycle delay of 35 s. The CP typically uses a 1 ms Hartmann-Hahn contact, with a centerband match of 50 kHz for $^1$H and $^{13}$C channels. The stepwise spectral filtration of biomolecules using the dynamical gradient was shown in Supplementary Fig. 8.

The narrow $^{13}$C peak linewidths of 0.4-1.0 ppm allowed us to unambiguously identify the signals of major polysaccharides. To resolve and assign the $^{13}$C signals of polysaccharides and proteins, 2D $^{13}$C-$^{13}$C correlation experiments were conducted. The 2D DP refocused *J*-INADEQUATE experiment[42] correlates the double-quantum (DQ) chemical shift, the sum of the two directly bonded $^{13}$C spins, with single quantum (SQ) chemical shifts. The experiment using DP, $^{13}$C-$^{13}$C J-coupling, and 1.7 s recycle delays preferentially detects mobile molecules, while the CP-based analog detects rigid molecules. The $^{13}$C-$^{13}$C intramolecular interactions were probed using a 100 ms dipolar-assisted rotational resonance (DARR) scheme. Long-range intermolecular cross-peaks were detected using a 1.5 s proton-driven spin diffusion (PDSD) experiment. The resolved chemical shifts were compared with the values indexed in the Complex Carbohydrate Magnetic Resonance Database (CCMRD; www.ccmrd.org)[81] to validate the chemical nature of the carbohydrates. The confirmed resonance assignments are listed in Supplementary Table 10.

Protein secondary structure was determined by the chemical shift differences between the observed $^{13}$C chemical shifts of Cα and the standard values of random-coil conformation[52]. The chemical shifts were obtained using 2D DP refocused *J*-INADEQUATE spectra for mobile amino acid residues and using 2D $^{13}$C-$^{13}$C DARR spectra for rigid proteins.

## Estimation of carbohydrate composition

We analyzed the peak volumes in 2D $^{13}$C-$^{13}$C spectra measured using 100 ms DARR and DP refocused *J*-INADEUQTAE schemes to estimate the composition of the rigid and mobile polysaccharides, respectively (Supplementary Table 2). The integration function of the Bruker Topspin software was used to get the peak volumes in 2D spectra. To minimize uncertainty caused by spectral crowding, only well-resolved signals were used for compositional analysis. The NMR peaks used for quantification, their resonance assignments, and the corresponding peak volumes, were provided in Source Data file.

## Solid-state NMR analysis of lipids

To probe phospholipid signals in membranes, 2D $^1$H-$^{13}$C refocused INEPT spectra were collected. This spectroscopic method has been applied previously to investigate the lipids in *Cryptococcus neoformans* cell walls[56]. This experiment is based on through-bond $^1$H-$^{13}$C magnetization transfer[55]. The two spin echoes contain two delays set to $1/4J_{CH}$ followed by another two delays set to $1/6J_{CH}$, which were calculated using a CH J-coupling of 140 Hz for carbohydrates. In solid samples, only the most mobile molecules with long transverse relaxation times could be observed using this experimental scheme. Therefore, the intrinsically dynamic lipids were efficiently detected. In addition, model phospholipids POPC and POPG (Avanti Polar Lipids) were measured for comparison. Around 50 mg of samples were packed into a 4 mm rotor. 1D $^{13}$C DP experiments (with a recycle delay of 3 s) and 2D $^1$H-$^{13}$C refocused INEPT experiments were conducted on both model lipid samples on a 400 MHz NMR spectrometer.

## Measurements of water contact and polymer dynamics

To examine the site-specific water contacts of polysaccharides and proteins, 1D and 2D water-edited $^{13}$C experiments were conducted[50,82], and such methods have been applied to understand the hydration profile of fungal and plant cell walls[35,51]. Briefly, a $^1$H-$T_2$ relaxation filter (1.2 ms × 2) was used to suppress the polysaccharide signals to less than 5%, while retaining 80% of water magnetization as shown in Supplementary Fig. 6. The water $^1$H polarization was then transferred to spatially proximal biomolecules through a $^1$H-$^1$H mixing period before transferring it to carbon via a 1-ms CP for high-resolution $^{13}$C detection. The $^1$H mixing time ranged from 0 ms to 100 ms for measuring 1D spectra and was fixed to 4 ms when the 2D spectrum was measured. Data obtained from the 1D spectra were analyzed by

plotting the relative intensities as a function of the square root of the ${}^1$H mixing time, which gave a buildup curve of peak intensity. The data obtained from the 2D scheme were analyzed by comparing the intensities between the water-edited spectrum (S) and the non-edited control spectrum ($S_0$), for each resolved carbon site. These $S/S_0$ intensity ratios reflect the extent of water retention around different carbon sites, which were documented in Supplementary Tables 4 and 6 for polysaccharides and proteins.

${}^{13}$C-$T_1$ relaxation was measured using CP-based Torchia $T_1$ scheme[83], with the z-filter duration varying from 0.1 μs to 8 s to provide complete relaxation curves as shown in Supplementary Fig. 7. ${}^{13}$C-detected ${}^1$H-$T_{1\rho}$ relaxation were measured using the Lee-Goldburg spin-lock sequence in which ${}^1$H spin diffusion was suppressed during both the spin-lock period and the CP period to obtain site-specific ${}^1$H relaxation information for protons that are directly bonded to a carbon site. A single exponential function was used to fit the data of both ${}^{13}$C-$T_1$ and ${}^1$H-$T_{1\rho}$ to obtain relaxation time constants, which are documented in Supplementary Table 5. All the spectra were analyzed using Topspin 4.0.8 and all the graphs were generated through OriginPro 2021b. All illustrative figures were prepared using Adobe Illustrator Cs6 V16.0.0.

### Preparation and experiments of other *Aspergillus* species

To compare with the *A. sydowii* sample (strain EXF-12860), uniformly ${}^{13}$C,${}^{15}$N-labeled mycelia were also prepared for two other *Aspergillus* halophilic fungal species including *A. atacamensis* (strain EXF-6660, isolated from a wall biofilm from a salt-water-exposed cave about 106 km south of Iquique city in hyperarid Acatama Desert in Chile)[84] and *A. destruens* (strain EXF-10411, isolated form canvass of oil painting in Slovenia)[85]. 20 g/L of ${}^{13}$C-glucose (Catalogue # CLM-1396-PK, Cambridge Isotope Laboratories, Inc.) and 2 g/L of ${}^{15}$N-labeled NH$_4$NO$_3$ (Catalogue # 366528, Millipore Sigma) were added to 100 mL mineral base media (Supplementary Table 8), which were then incubated for seven days at 28 °C and 200 rpm. Each strain was exposed to its optimal NaCl concentration: 1.0 M for *A. atacamensis* and 1.9 M for *A. destruens* as reported before[75,84]. 1D CP ${}^{13}$C and 2D ${}^{13}$C-${}^{13}$C 100 ms DARR experiments were conducted on a Varian VNMRS 850 MHz (19.9 Tesla) spectrometer at 13 kHz MAS at 290 K.

In parallel, three strains of the non-halophilic fungus *Aspergillus fumigatus* (Ku80, Af293, and RL578)[35,36] were cultured in 100 mL minimal liquid media by adding 10 g/L of ${}^{13}$C-glucose and 6.0 g/L of sodium nitrate for Ku80 and Af293, respectively, and 30 g/L of 13C-surcose and 2.0 g/L of sodium nitrate for RL578. The Ku80 sample was then incubated at 37 °C for 36 h under 200 rpm shaking. The Af293 culture was grown at 30 °C for 3d under 210 rpm shaking, and the RL578 sample was incubated at 30 °C under static condition. The composition of the medium used for each sample varied significantly, which has been documented in Supplementary Table 1. Approximately 50 mg of sample was used for ssNMR studies. 1D CP and 2D ${}^{13}$C-${}^{13}$C 53 ms CORD experiments[79] of *A. fumigatus* were conducted on a 400 MHz (9.4 Tesla) and an 800 MHz (18.8 Tesla) Brucker spectrometer respectively. The temperature was set to 293 K and the MAS frequency was 10–13.5 kHz for these experiments.

### Reporting summary

Further information on research design is available in the Nature Portfolio Reporting Summary linked to this article.

## Data availability

The unprocessed ssNMR data files generated in this study have been deposited in the Zenodo repository: https://doi.org/10.5281/zenodo.10001628. All relevant data that support the findings of this study are provided in the article and supplementary Information. The resonance assignment documented in Supplementary Table 10 was confirmed by cross-checking data available at CCMRD (publicly available at www.ccmrd.org). The source data underlying Figs. 3a, d, 4c–e, and 5c are provided as Source Data file. Source data are provided in this paper.

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

## Acknowledgements

This work was supported by the National Institutes of Health grant AI173270 to T.W. and Project-Conacyt-CB-285816 to R.A.B-G. N. G-C. acknowledge the support by the Slovenian Research Agency to Infrastructural Center Mycosmo (MRIC UL, I0-0022) and program P4-0432. The high-field NMR spectra were collected at the Environmental Molecular Sciences Laboratory (grid.436923.9), a DOE Office of Science scientific user facility sponsored by the Department of Energy's Office of Biological and Environmental Research and located at PNNL under contract DE-AC05-76RL01830. Y.P-L. and L.M-A. received postdoc and PhD fellowships from CONACyT. R.A.B-G acknowledges a Sabbatical fellowship (CVU 389816) from CONACyT. The NMR experiments of *A. fumigatus* samples were performed at the National High Magnetic Field Laboratory, which is supported by National Science Foundation Cooperative Agreement No. DMR-1644779 and the State of Florida.

## Author contributions

L.D.F., A.J., and L.M.-A. prepared 13C, 15N-labeled fungal cultures. L.D.F., A.S.L., M.D.W., and A.J. conducted NMR experiments. L.D.F. and M.D.W. analyzed the NMR data. J.-P.L., R.A.B-G., N.G-C., Y.P.L., L.D.F., and T.W. interpreted the structural data. L.D.F. and M.D.W. collected the TEM images. T.W. and R.A.B-G. designed and supervised the project. All authors contributed to the manuscript writing.

## Competing interests

The authors declare no competing interests.
