## [Peer Review File · Nature Communications]

Structural Adaptation of Fungal Cell Wall in Hypersaline EnvironmentREVIEWER COMMENTS

Reviewer #1 (Remarks to the Author):

General comments:

The authors of „Structural Adaptation of Fungal Cell Wall in Hypersaline Environment“ present a thorough dataset dealing with the halophilic fungi *Aspergillus sydowii* and its cell wall amendments between the *A. sydowii*'s optimal NaCl concentration, hypersaline and salt-deprived conditions. These changes were explored using solid-state NMR spectroscopy and TEM. The data is very interesting and are a very good contribution to the field.

Overall, the manuscript is well structured, and experiments were carried out and described in a thorough matter. The method used, ssNMR was previously applied by the same last author on *Aspergillus fumigatus* (<https://www.nature.com/articles/s41467-021-26749-z>). Although it is a different set-up looking at salt-adaption in the cell wall, I do miss a clear differentiation in the manuscripts: i) in the material and methods sections it is not referenced to the already published paper and ii) the authors could better describe in the current manuscript if there were any further developments and down-sides to the method. Further, in various parts of the manuscript it is not clear what information is already published in previous papers / is previous knowledge and what information is clearly new knowledge from the presented data. This is especially the case in the Discussion. In the Abstract the work on other halophilic and halotolerant organisms is mentioned which is also in the Materials and Methods section and beginning of Result section but more information/results are lacking for all further experiments .I would therefore recommend to either add more information on these two other fungi and their results or not mention it that prominent in the Abstract while it is actually not further discussed in the rest of the manuscript. Next to highlighting what is new knowledge, a deeper discussion of how the results fit into already known fungal salt adaptation strategies as well as a few more sentences concerning the application of the knowledge is lacking in the discussion section. Further it is not clear for every experiment if replicates were used and when.

More detailed comments

Abstract:

L27: what are harsh industrial processes, specify

L30: how many additional fungi – add number or names; see comment above

Introduction:

A few more information about what is known and what is the advantage of using ssNMR would be interesting. Further, how rigid vs mobile fraction were defined (literature?) would be helpful.

L58: reference 10 is a review paper and therefore the original paper reporting the results should be mentioned instead.

L70: The upregulation of ...: Reference is missing, explain better why these play a role in the cell wall

L92-94: What about the other halophiles/tolerant tested? Did you test the optimal salt concentration yourself or is this from a Reference? If second – add Reference; Same for the following sentences.

Results: For readability it would be helpful if either the colours used in the figures e.g. chti (orange) are mentioned in the text or mentioned in the figure itself – otherwise when reading the result in the text the reader has to jump to the figure description and then back to the figure etc.

L109: environment: give examples how and which environmental changes.

Fig. 1: change order of GM and GAG in either the figure or figure description.

L150: Explain in more detail why it could be a low population.

L158: Reference 37 is a review paper - refer to original results

L160: Rewrite sentence to make clear what are results from this study and what is from literature.

L184: ..the enrichment of GalN – here it is not clear which form GalN is meant and the description could be more detailed

184-187: This information would also fit well into the discussion section instead of results

Figure2: a) is this based on the TEM pictures? b) did you measure replicates?

Figure 3a and Supplementary Fig. 4: for comparison reasons it would be great to have all Intermolecular crosspeaks of the different salt concentrations at the same timepoint [s] – otherwise the result-description is hard to follow ; 3c: it would increase readability to mention the meaning of the colours also in the diagram or text

L223: consider using the word “conserved” here – as you are only looking at one strain

L232: in this section of results, it is hard to follow what was previously measured, what is measured within in this dataset, what is literature – please try to rephrase and move discussion points into discussion

L249: A reference for the stress condition is needed otherwise explain how it was defined as a stress condition in your setup.

L276: These results, especially concerning the hydrophobin should be described more detailed and moved to the discussion as it is a very important mechanism in halophilic fungi

Discussion

As mentioned above the Discussion should be rewritten to explain in a better way what is new, how this dataset improves knowledge about halophilic/tolerant adaptation mechanisms in fungi, compare to already known mechanisms and literature

L355: "should" this wording doesn't make sense to me in that case; is this a reference to literature?
Please rephrase

L381: On which results are these effects based on, describe in better detail.

389: "contradiction": discuss where these discrepancies in results might results from

L395: Reference missing

L404: describe in more detail how your results are helpful and in for which application.

Materials and Methods

companies' names for reagents, consumables, instruments are not mentioned + Time for the TEM fixation steps are missing

Statistics – how did you generate your graphs, which programs were used

L407: where did you get your strains from? Did you purchase them?

For all experiments: make clear how many replicates were used /measured and also refer to the measured replicates in the figure descriptions - It is mentioned for a few but not in all descriptions.

L407: what does "in presence" mean -the NaCl was added to the liquid PDA?

L408: Please add a reference concerning the optimal concentration or otherwise make clear how it was defined by your lab

L410: did you measure the amount of inoculum used somehow?

L420: where did these strains come from and how did you make sure the cells were treated equally concerning 13C, 15N?

L422: why was a different rpm used compared to the *Aspergillus sydowii*?

L423: which NaCl concentrations were used?

TEM: did you develop a new protocol or is this based on a previously published one – add Reference

ssNMR: Is this the same protocol as in your previous publication or were there any changes? Reference the publication if it was based on it.

Supplementary Material:

Supplementary Figure2: in the whole figure it is not clear which TEM picture is from which salt concentration; Description and legend should be improved.

Supplementary Figure 10: better describe what is what, especially in 10c/10d description are missing to know from which experiment/ NaCl concentration the different spectra are derived from

The supplementary Excel file (source data) starts with numbered TAB 2 – shouldn't it start with 1? Maybe an overview table describing what is to be found in each tab would help. Further I would recommend formatting the Excel file for better readability.

Reviewer #2 (Remarks to the Author):

This study by Fernando et al reports the biochemical characterization of the cell wall of one *Aspergillus* fungus in a particular environmental condition (in hypersaline environment). The authors used NMR spectroscopy, together with electron microscopy, to describe morphological changes of the cell envelope of *Aspergillus sydowii* and modifications of polysaccharide levels.

They report an extensive list of physico-chemical parameters (molar ratio, water accessibility, mobility, spatial packing) related to polysaccharides found in the cell envelope, and that are affected by the hypersaline environment used during the culture. The discussion of the results is rather short and mostly descriptive.

My main concerns are:(1) The study proposes to answer a very focused question related to fungal biology: is the growth of *Aspergillus sydowii* in hypersaline environment modify its cell wall organization. This aspect was already studied by one of the co-authors here: <https://doi.org/10.3390/cells9030525>. This publication already reported a morphological and biochemical characterization of *Aspergillus sydowii* growth in such hypersaline conditions. e.g. figure 4 panels C and D of <https://doi.org/10.3390/cells9030525> reports the change of the cell wall thickness in 2M NaCl vs. no NaCl and the ultrastructural analysis of fungal cross-sections, which in turn look very similar to what is presented in Figure S12 and figure 2 panel A of the current manuscript. Several claims of the manuscript look also very trivial (page 13: "rigidification and dehydration as observed in the cell wall polysaccharides" in hypersaline conditions). whether published in Nat. Comm. or elsewhere, I suggest the authors to make clear what are the new results compared to this previous publication.

(2) The manuscript proposes a description of the main polysaccharides found in the cell wall of *Aspergillus sydowii* (beta-glucans, chitin/chitosan, galactomannan and GAG), which by itself isn't very original since the composition of the cell wall in *Aspergillus* species is rather similar and has been described in details for *A. niger* or *A. fumigatus* by NMR (by the Tuo Wang group) and mass spectrometry by other groups after isolation of particular cell wall fractions.

(3) the topic is highly specific, and could be mostly of interest for the fungal cell wall community. The largely descriptive nature of the results could be difficult to understand for non-specialist readers of Nature Communication. Maybe a more specialized journal (J. Fungi, Biomacromolecules) is more suitable.

Responses to Reviewers

Summary of the revision: We would like to thank both reviewers for the highly constructive suggestions. We have diligently addressed all the comments in a point-to-point manner. To highlight the new findings, we have clearly distinguished them from previously reported results. The Discussion section has been fully rewritten, with more than doubled length (wordcount: 800 to 2 K), to cover eight important aspects related to structural discoveries and their implications. A new Figure 2 has been included to compare the halophilic species studied here with previously reported *A. fumigatus*. To demonstrate reproducibility, a new Supplementary Fig. 3 has been added, showing consistent results across three batches of replicates across all regimes of dynamics. We have provided more comprehensive experimental details by expanding the Methods section from 1.4 K words to 2.1 K words. To acknowledge the relevant literature and support our discussion, we have included more than twenty new references. We hope that these substantial changes have enhanced the clarity, readability, and significance of the new findings. Furthermore, we have deposited all original NMR data files in a publicly accessible repository. Two versions of the revised manuscript, one with changes tracked and the other with changes highlighted, are submitted to facilitate the review process. *Note: all the line numbers mentioned below only refer to the version with changes highlighted in blue.*

Reviewer #1:

General comments:

The authors of “Structural Adaptation of Fungal Cell Wall in Hypersaline Environment” present a thorough dataset dealing with the halophilic fungi *Aspergillus sydowii* and its cell wall amendments between the *A. sydowii*'s optimal NaCl concentration, hypersaline and salt-deprived conditions. These changes were explored using solid-state NMR spectroscopy and TEM. The data is very interesting and are a very good contribution to the field.

We would like to thank the reviewer for the encouraging comments on this study.

Overall, the manuscript is well structured, and experiments were carried out and described in a thorough matter. The method used, ssNMR was previously applied by the same last author on *Aspergillus fumigatus* (<https://www.nature.com/articles/s41467-021-26749-z>). Although it is a different set-up looking at salt-adaption in the cell wall, I do miss a clear differentiation in the manuscripts: i) in the material and methods sections it is not referenced to the already published paper and ii) the authors could better describe in the current manuscript if there were any further developments and down-sides to the method. Further, in various parts of the manuscript it is not clear what information is already published in previous papers / is previous knowledge and what information is clearly new knowledge from the presented data. This is especially the case in the Discussion. In the Abstract the work on other halophilic and halotolerant organisms is mentioned which is also in the Materials and Methods section and beginning of Result section but more

information/results are lacking for all further experiments. I would therefore recommend to either add more information on these two other fungi and their results or not mention it that prominent in the Abstract while it is actually not further discussed in the rest of the manuscript.

Thank you for the comments, we have now addressed all the concerns in each section. While the more detailed, point-to-point responses are provided in the later pages of this response letter, below are brief responses to the key points here:

“i) in the material and methods sections, it is not referenced to the recently published paper.”

We have now cited these references in the Methods sections. We also provided more information on how the culture condition and experiments are following previously established protocols. In addition, we have endeavored to improve the references throughout the main text by adding 24 new references, and by better referring to recently published papers at appropriate positions.

“ii) the authors could better describe in the current manuscript if there were any further developments and down-sides to the method.”

The NMR methods used in this study are a combination of those used in multiple previous studies of fungal materials in our group and in other groups (e.g., the INEPT experiment used by Dr. Stark to characterize lipids in *Cryptococcus*). We have now provided a summary of experimental approach in the Methods section and pointed out that the current limitation caused by the lengthy experimental time can be alleviated by recent development of proton detection and DNP methods for fungal analysis (**Lines 637-645**). In addition, we have now provided a brief introduction of the solid-state NMR capability in the Introduction (**Lines 78-87**).

“iii) Further, in various parts of the manuscript it is not clear what information is already published in previous papers / is previous knowledge and what information is clearly new knowledge from the presented data. This is especially the case in the Discussion”

We have now fully delineated new findings from previous knowledge. This is done by 1) removing all discussion component from the Results section and reorganize them in the Discussion section, 2) cite reference for each statement of previous knowledge and/or clearly mention it as “previously reported”, 3) refer to the corresponding figure number when a new finding is mentioned or discussed (e.g. figure panel number cited throughout **Lines 413-432** and **521-570**), and 4) add paragraphs (**Lines 521-542**) in the Discussion to directly compare new findings of *A. sydowii* with previous studies of *A. fumigatus*, 5) add a new comparison of NMR findings of *A. sydowii* in this study with the transcriptomic results published in recent studies (**Lines 453-519**). These changes has led to the more frequent citations of related references and figure numbers throughout the text, as well as the significant expansion of the Discussion section (wordcount increased from 800 to 2K). We hope these changes can better differentiate new findings and prior knowledge, and further, better correlate these them.

“In the Abstract the work on other halophilic and halotolerant organisms is mentioned which is also in the Materials and Methods section and beginning of Result section but more information/results are lacking for all further experiments.”

Thanks. Now we have added a separate section in the Results (**Lines 193-224**) to describe the similarity of *A. sydowii* with two other halophilic Aspergillus species. We also compared *A. sydowii* with *A. fumigatus*. A new **Figure 2** is also added to support the statements. The culture and experimental details are now provided as a dedicated sub-section in the Methods section (**Lines 736-755**).

Next to highlighting what is new knowledge, a deeper discussion of how the results fit into already known fungal salt adaptation strategies as well as a few more sentences concerning the application of the knowledge is lacking in the discussion section. Further it is not clear for every experiment if replicates were used and when.

This is a very helpful comment. We have now compared NMR results with already known fungal salt adaptation strategies by transcriptomic data, pointed out the consistency and discrepancy, and discussed the new insights brought in by our study (**Lines 453-519** and **544-557**). We also provided a more detailed writeup of the implications of the structural findings on the applications of halophilic fungi (**Lines 574-582**).

Data on three batches of replicates are now provided as a new **Supplementary Fig. 3** and has been described in the maintext (**Lines 240-242** in Results and **Lines 603-608** in Methods).

More detailed comments

Abstract:

L27: what are harsh industrial processes, specify

We have now briefly mentioned bioremediation and fermentation under conditions unfavorable for microbial growth as two brief examples (**Line 29-30**). More detailed descriptions of the applications of halophilic fungi are provided at the beginning of the Introduction (**Lines 46-51**) and echoed in the last paragraph of Discussion (**Lines 573-577**).

L30: how many additional fungi – add number or names; see comment above

We have removed the “other halophilic and halotolerant fungi” phrase from the Abstract for better focus. Instead, we have now added a new **Figure 2** and two new paragraphs to the Results section (**Lines 193-198** and **213-244**) to describe the comparisons of *A. sydowii* with two other halophilic species (*A. atacamensis* and *A. destruens*), and with the non-halophilic *A. fumigatus*, respectively. The sample preparation and experiments for *A. atacamensis*, *A. destruens*, and *A. fumigatus* are

added to the Methods section (**Lines 736-755**). The strain number and detailed culture conditions are also provided.

Introduction:

A few more information about what is known and what is the advantage of using ssNMR would be interesting. Further, how rigid vs mobile fraction were defined (literature?) would be helpful.

Thanks for the helpful advice! We have dedicated a s paragraph (**Lines 77-88**) to the introduction of solid-state NMR technique, where we briefly summarized the advantages of using cellular samples and the differentiation of mobile and rigid components for general readers. Key references are also added.

In addition, the recent studies of *A. fumigatus* cell walls are now reorganized as a separate paragraph (**Lines 90-103**). We pointed out that the methodology applied on *A. fumigatus* inspired the current study that directly observed the structural responses in the cell walls of halophilic fungal species.

L58: reference 10 is a review paper and therefore the original paper reporting the results should be mentioned instead.

We have added two original papers as new references (number 13 and 14): Zajc et al. *Appl. Environ. Microbiol.* 80, 247-256 (2014) and Kogej et al. *Microbiology* 153, 4261-4273 (2007).

L70: The upregulation of ...: Reference is missing, explain better why these play a role in the cell wall

This sentence has been removed. Instead, we provided references and more detailed information on hydrophobins in the later sections of Results and Discussions (**Lines 381-383** and **Lines 548-555**), where the structure of hydrophobins and its contribution to cell adhesion and rigidity are briefly summarized and related to the NMR-observed changes in protein hydration and rigidity. References (#58, 59, 78, 15, and 17) are also provided in these new descriptions of hydrophobins.

Bayry, J. et al. *PLOS Pathog.* 8, e1002700 (2012).

Plemenitaš, *Front. Microbiol.* 5, 199 (2014).

Berger, *J. Biol. Eng.* 13, 10 (2019)

Zajc, J. et al. *BMC Genom.* 14, 617 (2013).

Pérez-Llano et al. *Cells* 9, 525 (2020).

L92-94: What about the other halophiles/tolerant tested? Did you test the optimal salt concentration yourself or is this from a Reference? If second – add Reference, Same for the following sentences.

The optimal salt concentrations of all the halophiles used in this study were published recently from the labs of the coauthors (Batista-Garcia and Gunde-Cimerman). We have now added two references (#39 and 40):

Martinelli, L. et al. *Extremophiles* 21, 755-773 (2017)

Gonzalez-Abradelo, D. et al. *Bioresour. Technol.* 279, 287-297 (2019).

The references of the optimal references are included in the Introduction (**Line 111**) and the methods section (**Lines 744-745**). Throughout the maintext, we now have references whenever the optimal/stressed conditions were mentioned. In addition, we specified the names of the other halophiles here in the introduction.

Results:

For readability it would be helpful if either the colors used in the figures e.g chitin (orange) are mentioned in the text or mentioned in the figure itself – otherwise when reading the result in the text the reader has to jump to the figure description and then back to the figure etc.

Thanks. This is helpful. We have now labeled the carbohydrates with corresponding color codes in **Figure 1e**, **Figure 3b,c,f**, and **Figure 4b-e** as well as SI figures. Full carbohydrate names and/or abbreviations are provided directly in these figures whenever space is available.

In addition, we have also improved the captions of Figure 1 (**Lines 169-174**) to mention the abbreviation and color code of carbohydrates.

L109: environment: give examples how and which environmental changes.

We have added examples of environmental changes (**Lines 127-130**): “Variations in the structural organization of the fungal cell wall are often associated with alternations in environmental factors such as pH, temperature, oxidative stress, carbon source, and salinity. These environmental changes have an impact on the signaling pathways within the fungus and induce changes in cell wall biogenesis and morphology.”

Six references are also added: Sherrington et al. *PLOS Pathog.* 13, e1006403 (2017); Ikezaki et al. *Med. Mycol. J.* 60, 29-37 (2019); Komalapriya et al. *PLoS One* 10, e0137750 (2015); Hopke et al. *Trends Microbiol.* 26, 284-295 (2018); Hall, *Mol. Microbiol.* 97, 7-17 (2015)

Fig. 1: change the order of GM and GAG in either the figure or figure description.

The order of GM and GAG has been swapped in the captions, now it is matching the order of the figure.

L150: Explain in more detail why it could be a low population.

Thanks for the important point. We have now added a brief explanation with a new reference (Fontaine et al. *J. Biol. Chem.* 2000): “Previous chemical results have demonstrated that the GM- β -1,3-glucan-chitin complex constitutes only approximately 7% of the total cell wall weight in *A. fumigatus*. Because *A. sydowii* and *A. fumigatus* are taxonomically close, we can expect a similarly low amount of this polysaccharide complex in the cell wall of *A. sydowii*.”

L158: Reference 37 is a review paper - refer to original results

We have added two references where the chemical shifts of model chitin and chitosan samples were reported: Saito et al. *Macromolecules* 2424-2430 (1987) and Tanner et al. *Macromolecules* 23, 3576-3583 (1990).

We also retained the previous reference 37 (now #52) Fernando et al. *Front. Mol. Biosci.* 8, 727053 (2021). It is an original research article documenting the chemical shift comparison of fungal chitin and comparing with literature-reported model chitin samples.

L160: Rewrite sentence to make clear what are results from this study and what is from literature.

We have now expanded this sentence into a short paragraph (**Lines 193-198**) to clearly describe the similarity in three halophilic *Aspergillus* species (*Aspergillus atacamensis*, *Aspergillus destruens*, and *A. sydowii*). This is a new piece of information. This paragraph is also followed by another new paragraph comparing *A. sydowii* and *A. fumigatus* to show the structural difference, which has not been reported either (**Lines 213-224**). A new **Figure 2** is added.

L184: ..the enrichment of GalN – here it is not clear which form GalN is meant and the description could be more detailed

We have now specified it as the cationic form GalNH^{3+} in **Lines 256-259**.

184-187: This information would also fit well into the discussion section instead of the results.

Thanks. We have now moved it to Discussion (**Lines 435-439**), and combined with the summary of saline-induced changes to cell wall surface.

Figure 2: a) is this based on the TEM pictures? b) did you measure replicates?

a) Figure 2 is now Figure 3 due to the insertion of a new figure before it. Yes, Fig. 3a is based on new TEM pictures of the same *A. sydowii* strain EXF-12860. The samples are now cultured in liquid medium instead of wheat straw. We mentioned this difference in **Lines 230-234** where we

also improved the description of the TEM results and better differentiated the new data and the previous results.

b) We measured three replicates for each of the three concentrations (so, in total 9 of ^{13}C , ^{15}N -labeled samples for *A. sydowii*). These fungal samples are highly reproducible not only for the composition but also for the polysaccharide dynamics. A new **Supplementary Figure 3** is added to show the data and the results of the replicates are now described in **Lines 240-242**.

Figure 3a and Supplementary Fig. 4: for comparison reasons it would be great to have all Intermolecular crosspeaks of the different salt concentrations at the same timepoint [s] – otherwise the result-description is hard to follow; 3c: it would increase readability to mention the meaning of the colors also in the diagram or text

Thanks for the helpful advice. We have now paired the two 1.5 s spectra of 0,5 M and 2.0 M NaCl samples in **Fig. 4a** (previously Fig. 3a), and the two 0.1 s spectra in **Supplementary Fig. 5**. These direct comparisons have improved the readability.

L223: consider using the word “conserved” here – as you are only looking at one strain

We changed it to “a feature consistently found in both 0.5 M and 2.0 M *A. sydowii* samples” in **Line 297**.

L232: in this section of results, it is hard to follow what was previously measured, what is measured within in this dataset, what is literature – please try to rephrase and move discussion points into discussion

Thanks for pointing out the issue. We have substantially shortened this whole section on polymer hydration and dynamics by removing all discussion points to the Discussion section. We have also rephrased sentences where ambiguity might have been present. This section (**Lines 360-346**) now only contains 4 paragraphs covering 1) the basics of water-accessibility measured by NMR, 2) polymer hydration and its dependence on salinity, 3) polymer dynamics in *A. sydowii*, and 4) non-directional changes observed across salt gradient. We hope it is now more accessible to our readers.

L249: A reference for the stress condition is needed otherwise explain how it was defined as a stress condition in your setup.

We have now included a reference for the stress condition: Rodriguez et al. *J. Fungi* 7, 414 (2021). Also, in previous studies, we demonstrated that the transcriptional responses of *A. sydowii* are comparable to 0 M and 2 M NaCl compared to the optimal growth concentration of 0.5 M NaCl as reported in Pérez-Llano et al. *Cells* 9, 525 (2020). Since this strain of *A. sydowii* grows optimally in a range of 0.5-1.0 M NaCl, we were able to distinguish, at least at the transcriptomic level,

between stress responses and halophily responses when the fungus was grown at different salt concentrations.

L276: These results, especially concerning the hydrophobin should be described more detailed and moved to the discussion as it is a very important mechanism in halophilic fungi

We have now moved it to Discussion and we have now provided a new paragraph to better discuss the NMR results of proteins, and explain the importance of hydrophobins (**Lines 544-557**).

Discussion

As mentioned above the Discussion should be rewritten to explain in a better way what is new, how this dataset improves knowledge about halophilic/tolerant adaption mechanisms in fungi, compare to already known mechanisms and literature

Thanks for the suggestion. We have fully reorganized and substantially expanded the Discussion by 2.5-fold. We now clearly emphasize the new findings by mentioning the NMR data or cite the corresponding figure panels. The previously reported information is now either mentioned as previously reported or directly coupled with references.

The Discussion now covers eight important structural and biochemical aspects: 1) NMR-derived structure of *A. sydowii* cell walls, 2) key findings on molecular changes induced by hypersalinity, 3) emphasis of the new NMR insight by direct comparison with previously published transcriptomic data, 4) the first NMR-assessment of chitosan's role in cell wall structuring, 5) emphasis of the function of α -glucan, 6) a paragraph on proteins including hydrophobins, 7) non-directional changes in physical properties, and 8) implications to the applications of hydrophilies.

L355: "should" this wording doesn't make sense to me in that case; is this a reference to literature? Please rephrase

Thanks, we have included the reference, Chakraborty, A. et al. *Nat. Commun.* 12, 6346 (2021). We also enriched this sentence by providing more information and a reference is added: "GM and GAG also covalently connect to structural proteins through linkers containing hydrophobic amino acid residues that were preserved in the alkali-insoluble fraction of the cell wall and vanished in GM- and GAG-deficient mutants." The sentence has also been relocated to the Introduction where *A. fumigatus* cell wall structure was described (**lines 98-100**), because it represents a recent report rather than a new finding in the current study.

L381: On which results are these effects based, describe in better detail.

We have rewritten this sentence (**Lines 544-546**) and cited the corresponding figures: "In *A. sydowii* mycelia obtained under hypersaline conditions, proteins and lipids also become rigidified and dehydrated (Figs. 2b and 4c), similar to the observed changes in cell wall polysaccharides."

389: “contradiction”: discuss where these discrepancies in results might results from

We agree with the reviewer that the discrepancy in α -glucan content is an important observation. We have now provided a paragraph to discuss the possible reason and implication of this difference in *A. fumigatus* and halophilic *Aspergillus* species (**Lines 529-542**). Specifically, we added a brief point that “This can be correlated with the low transcription of α -1,3-glucan synthase genes in *A. sydowii*, as reported recently, or it could be due to physiological responses that hinder α -1,3-glucan biogenesis.”

L395: Reference missing

We have now expanded the description of the pathogenicity of *A. sydowii* and *A. destruens* (**Lines 537-540**), and also added two references: Pennerman et al. *BMC Microbiol.* 20, 342 (2020) and Gonzalez-Abradelo et al. *Bioresour. Technol* 279, 287-297 (2019).

L404: describe in more detail how your results are helpful and in for which application.

Thanks for the insightful advice. We have now elaborated more on how the finding helps in future research and application (**Lines 572-582**).

Materials and Methods

companies’ names for reagents, consumables, and instruments are not mentioned + Time for the TEM fixation steps are missing

We have added the company names for the reagents and consumables as well as the instrumentations throughout the Method section. The TEM fixation time is now added to the protocol (**Lines 612-614**): “The fungal samples were fixed for 12 h at 4 °C using 2.5% glutaraldehyde and 2 % paraformaldehyde (Electron Microscopy Sciences, Hartfield, PA) in 0.1 M phosphate buffer (pH 7.4).…”

Statistics – how did you generate your graphs, which programs were used

Throughout the Methods section, now we have mentioned all the software and programs used to generate graphs and for analysis. The information is also included in the Reporting Summary file that goes along with the submission of the manuscript.

- The data collected on Topspin version 3.5 and 2.5 and OpenVNMRJ 2.1a
- The graphs were generated through OriginPro. 2019b and Origin 2021b.
- The NMR spectra were analyzed and processed using TopSpin 4.0.8
- Statistical analysis for TEM imaging was conducted in Microsoft Excel 365.

- The cell wall thickness from TEM images were obtained from ImageJ V1.8.0_172
- TEM images Samples were observed with a JEOL JEM-1400 TEM (Peabody, MA) with an accelerating voltage of 120 kV at varying magnifications and photographed with Gatan Orius SC 1000A camera.
- The figures and illustrations are drawn in Adobe Illustrator Cs6 V16.0.0

L407: where did you get your strains from? Did you purchase them?

All the strains were from our coauthors of this manuscript. We have now provided detailed information on the fungal strains, together with the key references.

Lines 585-588: “*A. sydowii* strain EXF-12860 was used as the primary model fungus in this study. It was isolated from solid fermentation of sugarcane bagasse⁶ and was obtained from the EX microbial Culture Collection of infrastructural Center Mycosmo (MRIC UL) at University of Ljubljana (Slovenia).”

Lines 736-747: “To compare with the *A. sydowii* sample (strain EXF-12860), uniformly ¹³C, ¹⁵N-labeled mycelia were also prepared for two other *Aspergillus* halophilic fungal species including *A. atacamensis* (strain EXF-6660, isolated from a wall biofilm from a salt-water-exposed cave about 106 km south of Iquique city in hyperarid Acataama Desert in Chile)³⁹ and *A. destruens* (strain EXF-10411, isolated form canvass of oil painting in Slovenia)⁸⁶.”

For all experiments: make clear how many replicates were used /measured and also refer to the measured replicates in the figure descriptions - It is mentioned for a few but not in all descriptions.

All 1D ¹³C experiments were performed on all three replicates at each of the three salt concentrations (in total 9 samples for *A. sydowii*). This is not feasible for the 2D experiments, which requires long time for measuring each spectrum, and even longer time for analysis. However, the identical spectral patterns across the three batches at each NaCl concentration, and across the dynamics gradient of polymers within each sample have been confirmed in the new **Supplementary Figure 3**. We have now clarified this in **Lines 648-652** and **606-608** in the Methods section.

L407: what does “in presence” mean -the NaCl was added to the liquid PDA?

Yes, we have updated the description to “Potato Dextrose Agar (Thermo Fisher Scientific) supplemented with 0.5 M NaCl.”

L408: Please add a reference concerning the optimal concentration or otherwise make clear how it was defined by your lab

The references are added to define the optimum and the stress concentrations for *A. sydowii*. In addition, we also added references throughout the text when the stressed and optimum concentrations were mentioned.

L410: did you measure the amount of inoculum used somehow?

Since these fungi are in form of mycelium / and conidia, cell counting, or OD measurements does not apply to the inoculum amount. But approximately we added same-sized diced agar pieces (or agar plug ~ 5 mm²) to the liquid culture media. This is now mentioned in the sample preparation method section. (**Line 595-596**).

L420: where did these strains come from and how did you make sure the cells were treated equally concerning 13C, 15N?

The strains are from the coauthor (Ex Microbial Culture Collection of the Infrastructural Centre Mycosmo (MRIC UL), University of Ljubljana Slovenia). We have now provided the details of these strains in **Lines 736-747 and 585-588**.

There is no bias on labeling. This is because the sole carbon and nitrogen sources in the liquid culture media are ¹³C-glucose and NH₄¹⁵NO₃. This is clarified in **Line 593**.

L422: why was a different rpm used compared to the *Aspergillus sydowii*?

A. sydowii produces an excess of spores at 200 rpm in liquid medium. A large number of spores will adhere to the flask and their excessive production changes the rheology of the culture remarkably. *A. sydowii* were cultured at slow rpm while the other halophiles do not have this issue; therefore, *A. atacamensis* and *A. destruens* were cultured at 200 rpm. This is now mentioned in **Lines 596-599**.

L423: which NaCl concentrations were used?

We have updated the description of the NaCl concentration: “Each strain was exposed to its optimal salt concentration: 1.0 M for *A. atacamensis* and 1.9 M for *A. destruens* as reported before.” References are also provided.

TEM: did you develop a new protocol or is this based on a previously published one – add Reference

This is based on a protocol used for our previous study and we have now added the reference.

ssNMR: Is this the same protocol as in your previous publication or were there any changes? Reference the publication if it was based on it.

The solid-state NMR experiments are largely based on previously reported protocols applied to *A. fumigatus* cell walls, but also with modifications (e.g., INEPT). Now in the Method section, we mentioned how we adapted the protocol to the current study and cited the respective literature (Lines 637-645 and Lines 696-698).

Supplementary Material:

Supplementary Figure2: in the whole figure it is not clear which TEM picture is from which salt concentration; the Description and legend should be improved.

Thanks for pointing out the issue. We now labeled the salt concentration in each panel of the figure (now as **Supplementary Fig. 1**). We also improved the description in the legend: “The three columns of figures from the left to the right show the images of *A. sydowii* samples cultured at 0 M, 0.5 M, and 2 M NaCl conditions.”

Supplementary Figure 10: better describe what is what, especially in 10c/10d description are missing to know from which experiment/ NaCl concentration the different spectra are derived from

We have now labeled the NaCl concentration and *A. sydowii* in the panels c-e of this figure (now as Supplementary Fig. 11) as well as in the legend. We have made a similar update for another SI figure (Supplementary Fig. 10b,c) to improve clarity.

The supplementary Excel file (source data) starts with numbered TAB 2 – shouldn't it start with 1? Maybe an overview table describing what is to be found in each tab would help. Further I would recommend formatting the Excel file for better readability.

The source data file underlying Figs. 3a, d, Figs. 4c-e, and Fig. 5c (e.g., bar/pie graphs and statistical plots) are provided. We have now edited with better descriptions in the tabs and reformatted to improve the readability. Figures of NMR spectra do not have source data, but the original topspin file is now available in public data repository with the access link provided in the Data Availability section.

Reviewer #2:

This study by Fernando et al reports the biochemical characterization of the cell wall of one *Aspergillus* fungus in particular environmental condition (in hypersaline environment). The authors used NMR spectroscopy, together with electron microscopy, to describe morphological changes of the cell envelope of *Aspergillus sydowii* and modifications of polysaccharide levels.

They report an extensive list of physico-chemical parameters (molar ratio, water accessibility, mobility, spatial packing) related to polysaccharides found in the cell envelope, and that are

affected by the hypersaline environment used during the culture. The discussion of the results is rather short and mostly descriptive.

We fully agree with the reviewer on the limitations of the Discussion. The original manuscript has part of the discussion content mixed with the results to give direct structural implications after describing spectroscopic findings. We have now delineated these two components and further introduced more components to strengthen the Discussion section (with a 2.5-fold increase in length). The discussion now uses 14 paragraphs to covers 8 important aspects related to cell wall structure and modification by hypersalinity (detailed in the response to the first question below). The results and discussions are also better supported by literature with many new references. The detailed changes are provided below. We hope these major changes helped to improve the clarity and significance of this study.

My main concerns are:

(1) The study proposes to answer a very focused question related to fungal biology: is the growth of *Aspergillus sydowii* in hypersaline environment modify its cell wall organization. This aspect was already studied by one of the co-authors here: <https://doi.org/10.3390/cells9030525>. This publication already reported a morphological and biochemical characterization of *Aspergillus sydowii* growth in such hypersaline conditions. e.g. figure 4 panels C and D of <https://doi.org/10.3390/cells9030525> reports the change of the cell wall thickness in 2M NaCl vs. no NaCl and the ultrastructural analysis of fungal cross-sections, which in turn look very similar to what is presented in Figure SI2 and figure 2 panel A of the current manuscript. Several claims of the manuscript look also very trivial (page 13: "rigidification and dehydration as observed in the cell wall polysaccharides" in hypersaline conditions). whether published in Nat. Comm. or elsewhere, I suggest the authors to make clear what are the new results compared to this previous publication.

Thank you for the helpful advice. The previous study (Perez-Llano et al. 2020) is based on the transcriptomic analysis which tells the transcriptional activity (coding or noncoding) on a targeted gene, which represents the variation in the expression of genes. Expression level of cell wall genes does not really correlate to the proteins and the enzymatic activity, though the Perez-Llano et al. 2020, shows the expression level of cell wall related genes, it did not provide the in-situ structure of the cell wall and its molecular level packing. The current study, however, provides direct and high-resolution structural information that directly informs us on the rearrangement of cell walls. This has now been clarified as seven new paragraphs of Discussion (**Lines 453-519**).

We have fully rewritten the Discussion section, which covers important conceptual advances using 14 paragraphs covering eight major aspects: 1) *A. sydowii* cell wall structure, 2) modification by hypersaline condition, 3) the need of ssNMR after previous transcriptomic study, 4) the first assessment of chitosan's role in cell wall structuring, 5) consistency and diversity of glucans across *Aspergillus* species, 6) implication on proteins (including hydrophobins) and lipids, 7) non-directional changes in physical properties, and 8) implication for applications.

We decoupled all new findings from previous results whenever possible throughout the maintext. The literature results are now referred directly as “previously reported” or directly referred to a reference. All the new findings are now better described in the Discussion, with the corresponding figures cited.

In addition, all the data presented in this manuscript are original. The TEM data on the new samples cultured using liquid medium is better described and compared with the data reported in recent studies on straw-grown fungus (e.g., **Lines 228-236**).

With these changes, we hope the new structural findings directly from this study is better emphasized and the significance and biochemical relevance of our data can be appreciated by a broader audience.

(2) The manuscript proposes a description of the main polysaccharides found in the cell wall of *Aspergillus sydowii* (beta-glucans, chitin/chitosan, galactomannan and GAG), which by itself isn't very original since the composition of the cell wall in *Aspergillus* species is rather similar and has been described in details for *A. niger* or *A. fumigatus* by NMR (by the Tuo Wang group) and mass spectrometry by other groups after isolation of particular cell wall fractions.

We agree with the reviewer that *A. fumigatus* and *A. sydowii* shared partial similarity in some polysaccharides (e.g., chitin and β -glucan, but not for α -glucan nor chitosan). We have added a new **Figure 2** to show that the composition is similar within different halophilic *Aspergillus* species but has significant difference from the previously studied but non-halophilic *A. fumigatus* (**Lines 192-224**). In addition, we have emphasized the importance of these differences in two new paragraphs added to the Discussion section (**Lines 521-542**).

In addition, we have emphasized that the major discovery of this study is the structural adaptation mechanism (**Lines 427-439**), which has not yet been reported on the molecular level and cannot be fully resolved using previously published transcriptomic data. We hope these changes help to highlight the uniqueness of this study.

(3) the topic is highly specific and could be mostly of interest for the fungal cell wall community. The largely descriptive nature of the results could be difficult to understand for non-specialist readers of Nature Communication. Maybe a more specialized journal (J. Fungi, Biomacromolecules) is more suitable.

This study provided an answer to a biologically important question on how the fungus remodels its cell wall to survive through unfavorable and hypersaline conditions and provides insight into stress handling by these microbes. An answer to this question will deepen our understanding of life in extreme environments and could also serve as the basis for engineering better microbes with improved capability of adaptation for various applications. This aspect is now better elaborated in

Lines 572-582), where we summarized the important applications of halophiles and how the structural mechanisms of haloadaptation could inform on the improvement of these process, to further broaden the scope of this study. For easy access to the key findings by non-specialist readers, we have been trying to avoid the use of jargon and try to include technical aspects primarily in the Methods section and SI with more details in this revision. We hope that the various elements included in this manuscript (fungal biology, environmental microbiology, carbohydrate chemistry, biophysics, structural biology, and even transcriptomics) could be of interest to a broad audience.

REVIEWER COMMENTS

Reviewer #1 (Remarks to the Author):

The authors of the study improved the manuscript thoroughly and addressed all previous comments. The figures are now very clear and concise. Nevertheless, the introduction and discussion could be clearer and need some more improvement as stated below in detail.

Abstract

L27: Remove which or rewrite sentence

Introduction

While the introduction improved a lot, the hypothesis and aims of the study could be clearer stated. There is a lot of information about recent results from *Aspergillus fumigatus* but the link between this information and what is hypothesized for *A. sydowii* could be better phrased. For example the first sentences of the result section highlight much better what is the aim of the study. What makes this study special and useful for others in the halophile/fungal community?

L91: In this sentence, the „of the same Trichocomaceae family,“ doesn't make sense – maybe add „of the same Trichocomaceae family as *A. sydowii*“ for clarification

L 105-123 In the last paragraph of the introduction although changed from the previous version, it is not clear that the control strains were used in this study or if this is just from literature. I would recommend to state it clearer that you used these strains and leave the reference to the optimal salt conditions to the material and method section if you are not mentioning results from the study.

Results

Present form of verbs is sometimes used when past tense should be used or the other way around. Although the result section improved concerning the previous version, parts could still be moved to the discussion rather than to the result section. For example, L159ff.

Discussion

L453ff Although the comparison with the transcriptomic part is very interesting it should be made clearer in the text which results are your results made in this study and compare them to the

transcriptomic results made in another study. Just by adding „in this study“ to the specific sentences as well as „in the previous study“ would already help. In general this section could be made clearer and needs some more work.

L539: i guess you mean coral reefs?

Reviewer #2 (Remarks to the Author):

The manuscript has been clearly improved, experimental details are better presented. The new information content compared to previous papers is now well presented.

I still have an issue with the comparison of *A. sydowii* vs *A. fumigatus* cell wall composition (Fig. 2). Could the differences be simply explained as the result of different culture conditions used? Culture conditions have indeed a potential impact of the resulting cell wall composition (molar ratio of polysaccharides).

Responses to Reviewers

Reviewer #1:

The authors of the study improved the manuscript thoroughly and addressed all previous comments. The figures are now very clear and concise. Nevertheless, the introduction and discussion could be clearer and need some more improvement as stated below in detail.

Thanks for the positive comments on our previous revision. We appreciate the advice of how to improve the manuscript further. All the recommended changes have been incorporated.

Note: In addition to the submitted revised manuscript, a change-highlighted version is also included in the submission to facilitate the review process.

Abstract

L27: Remove which or rewrite sentence

Thanks. We have corrected this sentence.

Introduction

While the introduction improved a lot, the hypothesis and aims of the study could be clearer stated. There is a lot of information about recent results from *Aspergillus fumigatus* but the link between this information and what is hypothesized for *A. sydowii* could be better phrased. For example the first sentences of the result section highlight much better what is the aim of the study. What makes this study special and useful for others in the halophile/fungal community?

In our revised Introduction, we have added a new paragraph (**Lines 101-115**) that establishes a stronger connection between the ongoing study on *A. sydowii* and the existing knowledge base on *A. fumigatus*. We also emphasized the importance of using *A. sydowii* to establish a paradigm for understanding the structural modification mechanism of halophiles. This addition also allows us to articulate our central hypothesis more explicitly. Furthermore, the initial two sentences of the Results section from the previous version of the manuscript have been integrated into this paragraph of Introduction, which allows us to better distinguish results from introductory information.

At the same time, we also emphasized in the Discussion that the presence of chitosan of *sydowii* should be associated with halophily in **Lines 549-553**.

L91: In this sentence, the “of the same Trichocomaceae family” doesn’t make sense – maybe add “of the same Trichocomaceae family as *A. sydowii*” for clarification

Thanks. We have now changed it to “the same *Trichocomaceae* family as *A. sydowii*.” in **Line 91**.

L 105-123 In the last paragraph of the introduction although changed from the previous version, it is not clear that the control strains were used in this study or if this is just from literature. I would

recommend to state it clearer that you used these strains and leave the reference to the optimal salt conditions to the material and method section if you are not mentioning results from the study.

This is great advice. In this paragraph, we have eliminated references to specific salt concentrations and literature. Additionally, we now explicitly mention that these strains were examined in this study. The revised paragraph exclusively summarizes the key results.

The removed portions from this paragraph are not displayed in the highlighted version of the manuscript (as it is a deletion) but are included below for your information:

“which were cultured under the respective optimal salt concentrations reported for these fungal species”; “under the optimal salt concentration reported for *A. sydowii*”; and “of this fungus”

Results

Present form of verbs is sometimes used when past tense should be used or the other way around. Although the result section improved concerning the previous version, parts could still be moved to the discussion rather than to the result section. For example, L159ff.

We have now moved this part to Discussion as a separate paragraph (**Lines 449-457**).

In addition, we also checked through the Results section to use past tense when an NMR/experimental observation is described (**Lines 142, 151, 200, 219, 221, 239, 243, 247, 263, 264, 305, and 352**).

Discussion

L453ff Although the comparison with the transcriptomic part is very interesting it should be made clearer in the text which results are your results made in this study and compare them to the transcriptomic results made in another study. Just by adding “in this study” to the specific sentences as well as “in the previous study” would already help. In general this section could be made clearer and needs some more work.

We appreciate this helpful suggestion. We have now specified, e.g., our NMR detected changes and observations in this study or previously reported data, whenever possible. These changes happen to **Lines 476, 478, 486, 488, 490, 492, 494, 497, 501, 505, and 514**. We agree that cross-comparing NMR-observed structural changes with previously reported transcriptomic data could be challenging, and we hope this could help to make this section clearer.

L539: i guess you mean coral reefs?

Corrected.

Reviewer #2

The manuscript has been clearly improved, experimental details are better presented. The new information content compared to previous papers is now well presented.

Thanks for the positive comments about the revision. We hope it can be more accessible to our readers with these changes.

I still have an issue with the comparison of *A. sydowii* vs *A. fumigatus* cell wall composition (Fig. 2). Could the differences be simply explained as the result of different culture conditions used? Culture conditions have indeed a potential impact of the resulting cell wall composition (molar ratio of polysaccharides).

We fully agree with the reviewer on the potential influence of culture conditions on composition, which was also the reason for adding Supplementary Fig. 1 during the previous revision. Now we have provided additional details about the culture conditions and clarified that the significant changes observed between the two *Aspergillus* species are not due to variations in media composition or culture conditions. The three *A. fumigatus* samples are prepared with highly diverse conditions, with very consistent spectral patterns. At the same time, the three *A. sydowii* samples were prepared across a wide range of salt concentration. These changes did not impact the substantial changes detected in NMR.

These clarifications are outlined in the new **Supplementary Table 1**, which outlines the culture conditions, as well as in **Lines 223-230**, where the *A. fumigatus* and *A. sydowii* comparison is discussed, and in **Lines 775-782**, where the culture conditions are briefly described.